# Finding NeMo 🐠: Localizing Neurons Responsible For Memorization in Diffusion Models

**Dominik Hintersdorf**[*1,2]    **Lukas Struppek**[* 1,2]
**Kristian Kersting**[1,2,3,4]    **Adam Dziedzic**[5]    **Franziska Boenisch**[5]
[1]German Research Center for Artificial Intelligence (DFKI)
[2]Computer Science Department, Technical University of Darmstadt
[3]Hessian Center for AI (Hessian.AI)
[4]Centre for Cognitive Science, Technical University of Darmstadt
[5]CISPA Helmholtz Center for Information Security

## Abstract

Diffusion models (DMs) produce very detailed and high-quality images. Their power results from extensive training on large amounts of data—usually scraped from the internet without proper attribution or consent from content creators. Unfortunately, this practice raises privacy and intellectual property concerns, as DMs can memorize and later reproduce their potentially sensitive or copyrighted training images at inference time. Prior efforts prevent this issue by either changing the input to the diffusion process, thereby preventing the DM from generating memorized samples during inference, or removing the memorized data from training altogether. While those are viable solutions when the DM is developed and deployed in a secure and constantly monitored environment, they hold the risk of adversaries circumventing the safeguards and are not effective when the DM itself is publicly released. To solve the problem, we introduce NEMO, the first method to localize memorization of individual data samples down to the level of neurons in DMs' cross-attention layers. Through our experiments, we make the intriguing finding that in many cases, *single neurons* are responsible for memorizing particular training samples. By deactivating these *memorization neurons*, we can avoid the replication of training data at inference time, increase the diversity in the generated outputs, and mitigate the leakage of private and copyrighted data. In this way, our NEMO contributes to a more responsible deployment of DMs.

## 1   Introduction

In recent years, diffusion models (DMs) have made remarkable advances in image generation. In particular, text-to-image DMs, such as Stable Diffusion [32], DALL-E [30], or Deep Floyd [38] enable the generation of complex images given a textual input prompt. Yet, DMs carry a significant risk to privacy and intellectual property, as the models have been shown to generate verbatim copies of their potentially sensitive or copyrighted training data at inference time [8, 36]. This ability has often been linked to their *memorization* of training data [48, 12, 1, 5]. Memorization in DMs recently received a lot of attention [16, 47, 49, 8], and several mitigations have been proposed [46, 31, 36]. Those mitigations usually focus on either identifying potentially highly memorized samples and excluding them from training, monitoring inference and preventing their generation, or altering the inputs to prevent the verbatim output of training data [31, 36, 46]. While mitigations that rely on preventing the generation of memorized samples are effective when the DM is developed and deployed in a secure

---

*equal contribution, corresponding authors: {hintersdorf, struppek}@cs.tu-darmstadt.de
Code: `https://github.com/ml-research/localizing_memorization_in_diffusion_models`

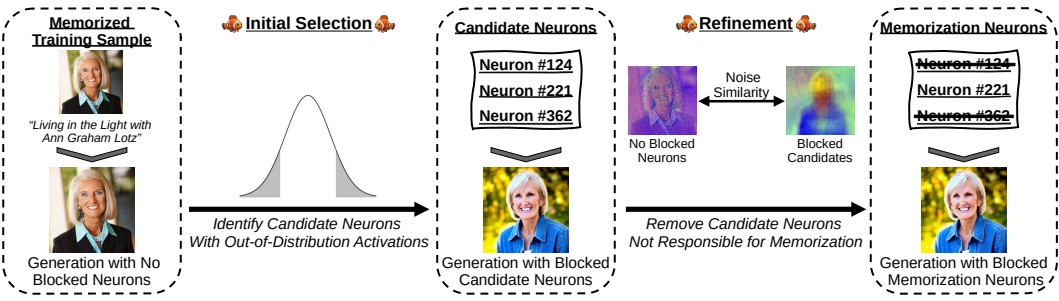

Figure 1: **Overview of NeMo**. For memorized prompts, we observe that the same (original training) image is constantly generated independently of the initial random seed. This yields severe privacy and copyright concerns. In the *initial stage*, NEMO 🐠 first identifies candidate neurons potentially responsible for the memorization based on out-of-distribution activations. In a *refinement* step, NEMO 🐠 detects the *memorization neurons* from the candidate set by leveraging the noise similarities during the first denoising step. Deactivating memorization neurons prevents unintended memorization behavior and induces diversity in the generated images.

environment, they hold the inherent risk of adversaries circumventing them. Additionally, they are not effective when the DMs are publicly released, such that users can freely interact with them.

As a first step to solving this problem, we propose FINDING **NE**URON ME**MO**RIZATION (NEMO), a new method for localizing where individual data samples are memorized inside the DMs. NEMO's localization tracks the memorization of training data samples down to the level of individual neurons in the DMs' cross-attention layers. To achieve this, NEMO relies on analyzing the different activation patterns of individual neurons on memorized and non-memorized data samples and identifying memorization neurons by outlier activation detection (as visualized in Fig. 2b). We empirically assess the success of NEMO on the publicly available DM Stable Diffusion [32]. Our findings indicate that most memorization happens in the value mappings of the cross-attention layers of DMs. Furthermore, they highlight that most training data samples are memorized by *just a few or even a single neuron*, which is surprising given the high resolution and complexity of the training data.

Based on the insights about where within the DMs individual data samples are memorized, we can prevent their verbatim output by deactivating the identified memorization neuron(s). We demonstrate the effect of NEMO in Fig. 1. Without our approach, the image generated for the memorized input prompt is the same, independent of the random seed for generation. By localizing the neuron responsible for the memorization through NEMO and deactivating it, we prevent the verbatim output of the training data and instead cause the generation of various non-memorized related samples. Hence, by relying on NEMO to localize and deactivate memorization neurons, we can limit memorization, which mitigates the privacy and copyright issues while keeping the overall performance intact.

In summary, we make the following contributions:

- We propose NEMO, the first method to localize where memorization happens within DMs down to the level of individual neurons.

- Our extensive empirical evaluation of localizing memorization within Stable Diffusion reveals that few or even single neurons are responsible for the memorization.

- We limit the memorization in DMs by deactivating the highly memorizing neurons and further show that this leads to a higher diversity in the generated outputs.

## 2 Background and Related Work

### 2.1 Text-to-Image Synthesis with Diffusion Models

Diffusion models (DMs) [37, 19] are generative models trained by progressively adding random Gaussian noise to training images and having the model learn to predict the added noise. After the training is finished, new samples can be generated by sampling an initial noise image $x_T \sim \mathcal{N}(\mathbf{0}, \mathbf{I})$ and then iteratively removing portions of the predicted noise $\epsilon_\theta(x_t, t, y)$ at each time step $t = T, \ldots, 1$.

This denoising process is formally defined by

$$x_{t-1} = \frac{1}{\sqrt{\alpha_t}} \left( x_t - \frac{1 - \alpha_t}{\sqrt{1 - \bar{\alpha}_t}} \epsilon_\theta(x_t, t, y) \right), \tag{1}$$

with variance scheduler $\beta_t \in (0, 1)$, $\alpha_t = 1 - \beta_t$ and $\bar{\alpha}_t = \prod_{i=1}^{t} \alpha_t$. The noise predictor $\epsilon_\theta(x_t, t, y)$, usually a U-Net [33], receives an additional input $y$ for conditional image generation.

Common text-to-image DMs [32, 30, 34] are conditioned on text embeddings $y$ computed by pre-trained text encoders like CLIP [29]. The typical way to incorporate the conditioning $y$ into the denoising process is the cross-attention mechanism [41]. (Cross-)Attention consists of three main components: query matrices $Q = z_t W_Q$, key matrices $K = y W_K$, and value matrices $V = y W_V$. All three matrices are computed by applying learned linear projections $W_Q, W_K$, and $W_V$ to the hidden image representation $z_t$ and the text embeddings $y$. The attention outputs are computed by

$$\text{Attention}(Q, K, V) = \text{softmax}\left( \frac{QK^T}{\sqrt{d}} \right) \cdot V, \tag{2}$$

with scaling factor $d$. Importantly, in most text-to-image models, the noise predictor receives guidance only through the cross-attention layers, which renders them particularly relevant for memorization.

## 2.2 Memorization in Deep Learning

**Memorization.** Memorization was extensively studied in supervised models and with respect to data labels [48, 1, 9]. Recently, studies have been extended to unlabeled self-supervised learning [25, 43]. In both setups, it was shown that memorization is required for generalization [12, 13, 43]. However, memorization also yields privacy risks [4, 7, 14, 40] since it can expose sensitive training data. In particular for generative models, including DMs, it was shown that memorization enables the extraction of training data points [4–6, 8, 36].

**Localizing Memorization.** Early work on localizing where inside machine learning (ML) models memorization happens focuses on small neural networks. Initial findings suggested that in supervised models, memorization happens in the deeper layers [2, 39]. However, more fine-grained analyses contradict these findings and identify that individual units, *i.e.,* individual neurons or convolutional channels throughout the entire model, are responsible for memorization [24]. To identify these, Maini et al. [24] deactivate units throughout the network until a label flip on the memorized training input image occurs. However, due to the unavailability of labels, this approach does not transfer to DMs.

**Memorization in Diffusion Models.** Recent empirical studies connect the model architecture, training data complexity, and the training procedure to the expected level of DM memorization [16], while others connect memorization to the generalization of the generation process [47]. Two types of memorization are usually distinguished: Verbatim memorization that replicates the training image exactly. And template memorization that reproduces the general composition of the training image while having some non-semantic variations at fixed image positions [45]. Existing approaches for detecting memorized training samples are based on statistical differences in the model behavior when queried with memorized prompts. These approaches explore differences in predicted noise magnitudes [46], the distribution of attention scores [31], the amount of noise modification in one-step synthesis [45], and the edge consistency in generated images [45]. Our work is orthogonal to these detection methods, focusing on the exact *localization of memorization* in the DM's U-Net rather than detecting memorized samples.

Previously proposed methods for mitigating memorization during inference either rescale the attention logits [31] or adjust the text embeddings with a gradient-based approach to minimize the magnitude of noise predictions [46]. However, these inference time mitigation strategies are easy to deactivate in practice and provide no permanent mitigation strategies for publicly released models. In contrast, related training-based mitigation strategies [46, 31] require re-training an already trained model like Stable Diffusion, which is time- and resource-intensive. We show that NEMO can reliably identify individual neurons responsible for memorizing specific training samples. Pruning these neurons effectively mitigates memorization, does not harm the general model performance, and provides a more permanent solution to avoid training data replication.

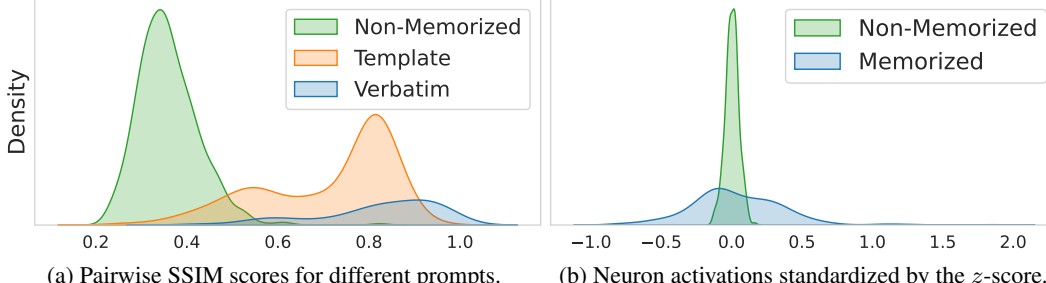

(a) Pairwise SSIM scores for different prompts.  (b) Neuron activations standardized by the $z$-score.

Figure 2: **Differences Between Memorized and Non-memorized Prompts. (a)** depicts the distribution of pairwise SSIM scores between initial noise differences starting from different seeds. Since the noise trajectories are more consistent for memorized samples, the score reflects the degree of memorization. **(b)** shows the distribution of the $z$-scores of each neuron in the first cross-attention value layer. Memorization neurons produce considerably higher activations, here depicted as standardized $z$-scores, for memorized prompts, allowing them to be identified by outlier detection.

## 3 NeMo: Localizing and Removing Memorization in Diffusion Models

NEMO, our method for detecting *memorization neurons*, consists of a two-step selection process:

**(1) Initial Selection**: We first identify a broad set of candidate neurons that might be responsible for memorizing a specific training sample. This initial selection is coarse-grained to speed up the computation among the many neurons in DMs. Consequently, it might select false positives, *i.e.,* neurons not directly responsible for memorization.

**(2) Refinement:** In the refinement step, we filter neurons out to reduce the size of the initial candidate set. After refinement, we deactivate the remaining *memorization neurons* to remove memorization.

In our study, we apply this two-step approach of NEMO to detect memorization neurons in the DM's cross-attention layers, the only components that directly process the text embeddings. Image editing research [17, 42, 10] shows that cross-attention layers highly influence the generated content, so we expect them to be the driving force behind memorization. We analyze the impact of blocking individual key and value layers in Appx. C.8 and the influence of blocking neurons in the convolutional layers in Appx. C.9. Our results show that the value layers in the down- and mid-blocks of the U-Net indeed have the highest memorization effect, whereas value layers in the up-blocks barely affect the memorization. Deactivating the outputs of neurons in value layers completely blocks the information flow of the guidance signal and, hence, potential memorization triggers. Deactivating the key layers in cross-attention also impacts memorization but often impedes the image-prompt alignment. Similarly, deactivating neurons in the convolutional layers of the U-Net did not mitigate memorization. Therefore, we limit our search on memorization neurons to the value layers of the U-Net's down- and mid-blocks. While the identified neurons effectively mitigate the data replication problem, we emphasize that other parts of the U-Net might also play a crucial role in memorizing training data. Specifically, the identified neurons trigger the data replication, which is then executed by other parts of the U-Net, such as convolutional and fully connected layers. Deactivating the memorization neurons in value layers effectively interrupts the memorization chain and replication process. Before detailing the two steps of NEMO's selection process in Sec. 3.2 and Sec. 3.3, we first introduce how we quantify memorization strength in the next section. We provide detailed algorithmic descriptions for each of NEMO's components in Appx. D.

### 3.1 Quantifying the Memorization Strength

Intuitively, the denoising process of DMs for memorized prompts follows a rather consistent trajectory to reconstruct the corresponding training image, yielding image generations with little diversity. Conversely, the denoising trajectory highly depends on the initially sampled noise for non-memorized prompts [46]. We measure the similarity between the first denoising steps for different initial seeds as a proxy to compare the denoising trajectories and to quantify the memorization strength. The higher the similarity, the more consistent the denoising trajectories, which indicates stronger memorization. Let $x_T \sim \mathcal{N}(\mathbf{0}, \mathbf{I})$ be the initial noise following the denoising process described in Sec. 2.1.

Let $\epsilon_\theta(x_T, T, y)$ further denote the initial noise prediction. We found that the normalized difference between the initial noise and the first noise prediction $\delta = \epsilon_\theta(x_T, T, y) - x_T$ for memorized prompts is more consistent for different seeds than for non-memorized prompts. We visualize this phenomenon for some initial noise differences in Appx. C.6.

To detect the grade of memorization, we, therefore, use the similarity between the noise differences $\delta^{(i)}$ and $\delta^{(j)}$ generated with seeds $i$ and $j$ as a proxy. We measure the similarity with the common structural similarity index measure (SSIM) [44]. A formal description of the SSIM $\in [-1, 1]$ score and an additional experiment outlining how the SSIM score can be used to detect memorization in the first place is provided in Appx. B.4.

A higher SSIM indicates higher similarity between the noise differences, reflecting a higher degree of memorization. Notably, the SSIM computation only requires a single denoising step per seed, which makes the process fast. To set a memorization threshold $\tau_{\mathrm{mem}}$, starting from which we define a sample as memorized, we first compute the mean SSIM on a holdout set of non-memorized prompts. We compute the pairwise SSIM between ten different initial noise samples for each prompt and take the maximum score. After that, we average the scores across all prompts and set the threshold $\tau_{\mathrm{mem}}$ to the mean plus one standard deviation. We consider the current image generation non-memorized if the maximum pairwise SSIM scores are below this threshold $\tau_{\mathrm{mem}}$. Fig. 2a shows the distribution of SSIM scores for memorized and non-memorized prompts, demonstrating that memorized prompts lead to a substantially higher score.

## 3.2 Initial Candidate Selection for Memorization Neurons

With our measure for quantifying the strength of memorization defined, we move on to detail the first step of our NEMO' localization. Our initial neuron selection procedure is based on the observation that activation patterns of memorized prompts differ from the ones of non-memorized prompts on the neuron level. Fig. 2b underlines this observation by plotting the standardized activation scores for memorized and non-memorized samples in the first value layer. Leveraging this insight, we identify memorization neurons as the ones that exhibit an out-of-distribution (OOD) activation behavior. We first compute the standard activation behavior of neurons on a separate hold-out set of non-memorized prompts. Then, we compare the activation pattern of the neurons for memorized prompts and identify neurons with OOD behavior. Let the cross-attention value layers of a DM be $l \in \{1, \dots, L\}$. We denote the activation of the $i$-th neuron in the $l$-th layer for prompt $y$ as $a_i^l(y)$. The activation values are averaged across the absolute neuron activations for each token vector in the text embedding. Let $\mu_i^l$ be the pre-computed mean activation and $\sigma_i^l$ the corresponding standard deviation for this neuron. To detect neurons potentially responsible for the memorization of a memorized prompt $y$, we compute the standardized $z$-score [20], defined as

$$z_i^l(y) = \frac{a_i^l(y) - \mu_i^l}{\sigma_i^l} . \tag{3}$$

The $z$-score quantifies the number of standard deviations $\sigma_i^l$ by which the activation $a_i^l(y)$ is above or below the mean activation $\mu_i^l$. Here, the activation $a_i^l(y)$ is calculated by taking the mean over the absolute token activations. To identify a neuron as exhibiting an OOD activation behavior, we set a threshold $\theta_{\mathrm{act}}$ and assume that neuron $i$ in layer $l$ has OOD behavior if $|z_i^l(y)| > \theta_{\mathrm{act}}$. The lower the threshold $\theta_{\mathrm{act}}$, the more neurons are labeled as OOD and added to the *memorization neuron* candidate set. An algorithmic description of the OOD detection step is provided in Alg. 2 in Appx. D.2.

Fig. 2a shows that the pairwise SSIM score can be used to measure the generated sample's degree of memorization. Hence, to get an initial selection of *memorization neurons*, we calculate the standardized $z$-scores for all neurons and start with a relatively high value of $\theta_{\mathrm{act}} = 5$. We deactivate all neurons with OOD activations given the current threshold $\theta_{\mathrm{act}}$, i.e., setting the output of a neuron to 0 if $|z_i^l(y)| > \theta_{\mathrm{act}}$, to reduce the memorization strength. If, after deactivating these neurons, the memorization score is not below the threshold $\tau_{\mathrm{mem}}$, we then iteratively decrease the activation threshold $\theta_{\mathrm{act}}$ by 0.25 and update the candidate set until the target memorization score $\tau_{\mathrm{mem}}$ is reached.

The activation patterns of some neurons in the network show high variance, even on non-memorized prompts. Such neurons can also be memorization neurons, but due to their high activation variance, they might not be detected by our OOD approach based solely on the $z$-scores. Therefore, we also add the top-$k$ neurons of each layer with the highest absolute activation on the memorized prompt $y$

to our current candidate set to account for such high-variance neurons. We start by setting $k = 0$ and increase $k$ at each iteration by one if the memorization score is still above the threshold $\tau_{\text{mem}}$. We detail our initial selection process in Alg. 3 in Appx. D.3. All neurons identified by our OOD approach and the neurons with the $k$ highest activations are then collected in the neuron set $S_{\text{initial}}$. Since not all neurons in set $S_{\text{initial}}$ might be memorization neurons, we refine this set in the next step.

### 3.3 Refinement of the Candidate Set

In this step, we take the set of identified neurons $S_{\text{refined}} = S_{\text{initial}}$ and remove the neurons that are actually not responsible for memorization. To speed up this process, we first group the identified neurons layer-wise, leading to the neuron set $S^l_{\text{refined}}$ for layer $l$. We iterate over the individual layers $l \in \{1, \ldots, L\}$ and re-activate all identified neurons $S^l_{\text{refined}}$ from a single layer $l$ while keeping the identified neurons in the remaining layers deactivated.

We then compute the SSIM-based memorization score and check if it is still below the threshold $\tau_{\text{mem}}$. If the memorization score does not increase above the threshold $\tau_{\text{mem}}$, we consider the candidate neurons $S^l_{\text{refined}}$ of layer $l$ as not memorizing and remove them from our set of neurons $S_{\text{refined}}$. After iterating over all layers, the set $S_{\text{refined}}$ only contains neurons from layers that substantially influence the memorization score.

Next, we individually check each remaining neuron in the set $S_{\text{refined}}$ by re-activating this particular neuron while keeping all other neurons in the set $S_{\text{refined}}$ deactivated. Again, if the memorization score computed on the remaining deactivated neurons does not exceed the memorization threshold $\tau_{\text{mem}}$, we remove this neuron from the set $S_{\text{refined}}$. After iterating over all neurons in $S_{\text{refined}}$, we consider the remaining neurons as memorizing and denote the final set of memorization neurons as $S_{\text{final}}$. We detail this refinement approach in Alg. 4 in Appx. D.4.

## 4 Experiments

We now empirically evaluate NEMO's localization in text-to-image DMs.

**Models and Datasets:** We follow current research on memorization in DMs [46, 31] and investigate memorization in Stable Diffusion v1.4 [32]. Our set of memorized prompts consists of 500 LAION prompts [35] provided by Wen et al. [46]. We analyzed the prompts using the Self-Supervised Descriptor (SSCD) score [28], a model designed to detect and quantify copying in DMs. The lower the score, the less similar the contents in the image pairs. Additionally, we split the dataset into verbatim-(*VM*) and template-memorized (*TM*) samples to enable a more detailed analysis of results. The hyperparameter selection and experimental conduction are independent of the type of memorization. If not further specified, we used the same hyperparameters for all the experiments in the paper.

Images generated by VM prompts match the training image exactly, *i.e.,* pixel-wise, independent of the chosen seed. TM prompts, on the other hand, reproduce the general composition of the training image while having some non-semantic variations at fixed image positions. Details about the analysis and the annotation can be found in Appx. C.1.

Other publicly available models, like Stable Diffusion v2 and Deep Floyd [38], are trained on more carefully curated and deduplicated datasets. We thoroughly checked for memorized prompts using the tools by Webster [45] and our SSIM-based memorization score but could not identify any properly memorized prompts. This result aligns with related research on memorization in DMs [46, 31].

**Metrics:** We split our metrics into memorization, diversity, and quality metrics. The memorization metrics measure the degree of **memorization** still present in the generated images. We generate ten images for each memorized prompt with activated/deactivated memorization neurons and measure the cosine similarities between image pairs using SSCD embeddings to quantify the memorization. We denote this metric by $\text{SSCD}_{\text{Gen}}$. Since the generated images without deactivated neurons also differ in their degree of memorization from the original training images, we additionally measure the degree of memorization towards the original training images and denote this metric as $\text{SSCD}_{\text{Orig}}$. Higher SSCD scores indicate a higher degree of memorization.

Our **diversity** metric assesses the variety of images generated for the same memorized prompt with different seeds. Diversity is usually low for memorized samples, and generated images almost always depict the same image. Deactivating memorization neurons increases the diversity in the generations.

Table 1: **Impact of Deactivating the Memorization Neurons.** Keeping all neurons active (1st row) and randomly deactivating neurons (3rd row) has no impact on memorization. However, deactivating the memorization neurons located by NEMO (8th row) successfully mitigates memorization, increases diversity, and maintains prompt alignment. These results are comparable to the gradient-based mitigation strategies adjusting the prompt embeddings (2nd row) and the attention scaling (4th row). Adding random tokens also reduces memorization. However, for 1 or 4 tokens, the memorization, as quantified by the SSCD scores, is still higher than with deactivated memorization neurons. Adding 10 random tokens leads to comparable mitigation but also reduces the prompt alignment score.

| Setting | Memorization Type | Deactivated Neurons | $\downarrow SSCD_{\text{Orig}}$ | $\downarrow SSCD_{\text{Gen}}$ | $\downarrow D_{\text{SSCD}}$ | $\uparrow A_{\text{CLIP}}$ |
|---|---|---|---|---|---|---|
| All Neurons Activate (Default) | Verbatim | 0 | $0.83 \pm 0.16$ | $1.0 \pm 0.0$ | $0.99 \pm 0.01$ | $0.32 \pm 0.02$ |
| | Template | 0 | $0.04 \pm 0.04$ | $1.0 \pm 0.0$ | $0.17 \pm 0.06$ | $0.31 \pm 0.02$ |
| Prompt Embedding Adjustment (Wen et al. [46]) | Verbatim | 0 | $0.04 \pm 0.02$ | $0.08 \pm 0.03$ | $0.08 \pm 0.03$ | $0.30 \pm 0.02$ |
| | Template | 0 | $0.03 \pm 0.02$ | $0.08 \pm 0.03$ | $0.09 \pm 0.03$ | $0.31 \pm 0.02$ |
| Deactivating Random Neurons | Verbatim | $4 \pm 3$ | $0.80 \pm 0.11$ | $0.999 \pm 0.0$ | $0.99 \pm 0.01$ | $0.32 \pm 0.02$ |
| | Template | $21 \pm 18$ | $0.05 \pm 0.04$ | $0.997 \pm 0.0$ | $0.16 \pm 0.06$ | $0.31 \pm 0.02$ |
| Attention Scaling (Ren et al. [31]) | Verbatim | 0 | $0.08 \pm 0.04$ | $0.14 \pm 0.07$ | $0.15 \pm 0.05$ | $0.31 \pm 0.02$ |
| | Template | 0 | $0.05 \pm 0.02$ | $0.19 \pm 0.12$ | $0.12 \pm 0.03$ | $0.31 \pm 0.02$ |
| Adding 1 Random Token (Somepalli et al. [36]) | Verbatim | 0 | $0.59 \pm 0.31$ | $0.68 \pm 0.31$ | $0.67 \pm 0.33$ | $0.31 \pm 0.02$ |
| | Template | 0 | $0.04 \pm 0.03$ | $0.16 \pm 0.05$ | $0.17 \pm 0.05$ | $0.31 \pm 0.02$ |
| Adding 4 Random Tokens (Somepalli et al. [36]) | Verbatim | 0 | $0.09 \pm 0.06$ | $0.12 \pm 0.09$ | $0.15 \pm 0.05$ | $0.30 \pm 0.02$ |
| | Template | 0 | $0.04 \pm 0.03$ | $0.13 \pm 0.04$ | $0.15 \pm 0.04$ | $0.30 \pm 0.02$ |
| Adding 10 Random Tokens (Somepalli et al. [36]) | Verbatim | 0 | $0.03 \pm 0.02$ | $0.07 \pm 0.05$ | $0.11 \pm 0.03$ | $0.28 \pm 0.03$ |
| | Template | 0 | $0.03 \pm 0.03$ | $0.08 \pm 0.05$ | $0.12 \pm 0.04$ | $0.29 \pm 0.03$ |
| **Deactivating Memorization Neurons (NEMO)** | **Verbatim** | $\mathbf{4 \pm 3}$ | $\mathbf{0.09 \pm 0.06}$ | $\mathbf{0.10 \pm 0.07}$ | $\mathbf{0.16 \pm 0.06}$ | $\mathbf{0.31 \pm 0.02}$ |
| | **Template** | $\mathbf{21 \pm 18}$ | $\mathbf{0.05 \pm 0.03}$ | $\mathbf{0.05 \pm 0.04}$ | $\mathbf{0.12 \pm 0.05}$ | $\mathbf{0.31 \pm 0.02}$ |

We quantify the sample diversity by computing the pairwise cosine similarity between the SSCD embeddings of different images generated for the same prompt. We refer to this metric as $D_{\text{SSCD}}$, for which lower values indicate more image diversity.

To assess the overall image **quality** of a DM with activated/deactivated neurons, we compute the Fréchet Inception Distance (FID) [18], the CLIP-FID [21], and the Kernel Inception Distance (KID) [3] on COCO [22] prompts. All quality computations follow Parmar et al. [26] to avoid biased results. Additionally, we compute the similarities between the generated images and the input prompts using CLIP scores [29] to ensure the alignment $A_{\text{CLIP}}$ between generated images and their prompts. The higher the alignment, the better the generated images represent the concepts described in the prompt.

Importantly, we use different seeds for detecting memorization neurons with NEMO and the metric computations to avoid undesired biases during the evaluation due to seed overfitting. We always state each metric's median value and absolute deviation across ten seeds, except the quality metrics (FID, CLIP-FID, KID, and $A_{\text{CLIP}}$), for which we used five different seeds.

**Memorization Threshold:** We set the memorization score threshold to $\tau_{\text{mem}} = 0.428$, which corresponds to the mean plus one standard deviation of the pairwise SSIM score between initial noise differences measured on a holdout dataset of 50,000 LAION [35] prompts.

**Baselines:** As a baseline, we repeated the image generations five times but replaced the deactivated memorization neurons with random neurons from the same layer. We also generated images using the inference mitigation strategy proposed by Wen et al. [46], which performs a gradient-based adjustment of the text embeddings. Importantly, gradient-based mitigation strategies are memory-intensive, particularly for larger batch sizes. NEMO, however, computes no gradients, which enables the approach to also work on machines with limited computing resources. Additionally, we also selected the attention scaling method by Ren et al. [31] and the addition of random tokens, proposed by Somepalli et al. [36], as baselines.

## 4.1 Localizing Memorization Down To Individual Neurons

We begin by demonstrating the effectiveness of our memorization localization method. Tab. 1 presents the quantitative results for images generated with the identified memorization neurons deactivated. NEMO detected a median of 4 and 21 memorization neurons for VM and TM prompts, respectively. For VM prompts, deactivating these memorization neurons significantly decreases memorization, as reflected by low memorization metrics $SSCD_{\text{Orig}}$ and $SSCD_{\text{Gen}}$, while increasing the image diversity in terms of pairwise similarity $D_{\text{SSCD}}$. However, the $SSCD_{\text{Orig}}$ does not change noticeably for TM prompts.

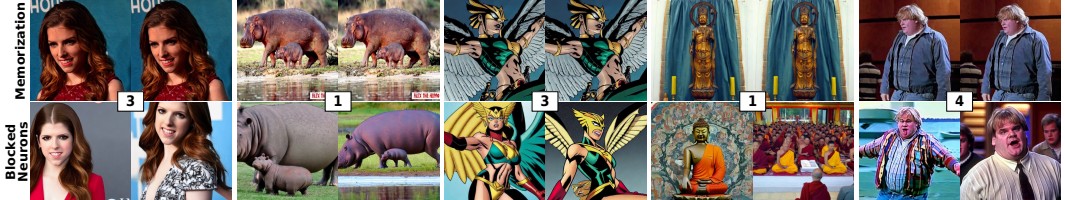

Figure 3: **Impact of Deactivating Memorization Neurons**. The top row shows images generated with memorized prompts, closely replicating the training images. The bottom row demonstrates that deactivating memorization neurons increases diversity and mitigates memorization. Notably, only a few neurons (counts indicated by digits in the boxes) are responsible for these memorizations.

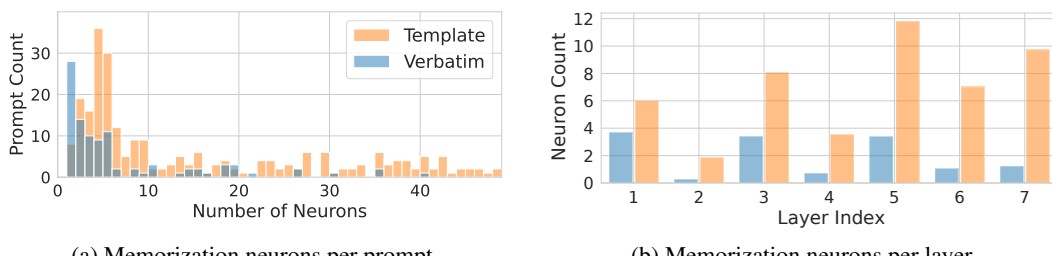

(a) Memorization neurons per prompt.

(b) Memorization neurons per layer.

Figure 4: **Distribution of Memorization Neurons.** **(a)** shows the number of prompts that are memorized by a fixed number of neurons, e.g., the verbatim memorization of 28 prompts is located in single neurons. **(b)** depicts the average number of memorization neurons per layer and prompt.

This behavior results from the fact that TM prompts typically memorize specific parts of the original training image, such as objects or compositions, rendering the $SSCD_{Orig}$ metric less informative. In contrast, the $SSCD_{Gen}$ score, which compares similarities between images generated with and without the deactivated neurons, provides a more accurate measure. This score highlights that deactivating the identified neurons effectively alters the images and mitigates memorization. Importantly, the image-prompt alignment $A_{CLIP}$ remains constant in all cases, indicating that deactivating memorization neurons does not result in misguided image generations. We visualize examples of deactivating memorization neurons to avoid data replication and increase diversity in Fig. 3.

Comparing the results of deactivating the neurons identified by NEMO with those obtained from randomly deactivated neurons highlights that only a specific subset of neurons is actually responsible for memorizing a prompt. While deactivating the identified memorization neurons significantly impacts both memorization and the diversity of the generated images, randomly deactivating neurons has no noticeable effect.

Moreover, the mitigation effect of deactivating memorization neurons is comparable to the state-of-the-art method of adjusting the prompt embeddings [46]. Yet, adjusting the prompt embeddings requires gradient computations for each seed and prompt, which are time- and memory-expensive, especially with large batch sizes. In contrast, once the memorization neurons are identified using our gradient-free NEMO, no additional computations are needed during image generations, thus adding no overhead to the generation process.

## 4.2 Analyzing the Distribution of Memorization Neurons

Next, we analyze the distribution of the memorization neurons. Fig. 4a shows the total number of neurons responsible for memorizing specific prompts. Typically, a small set of neurons is responsible for verbatim memorization. For instance, 28 VM prompts from our dataset are memorized by a *single neuron*. Additionally, five or fewer neurons replicate two-thirds of VM images, indicating that verbatim memorization can often be precisely localized within the model. Template memorization can also frequently be pinpointed to a small set of neurons, with about 30% of TV replication triggered by five or fewer neurons.

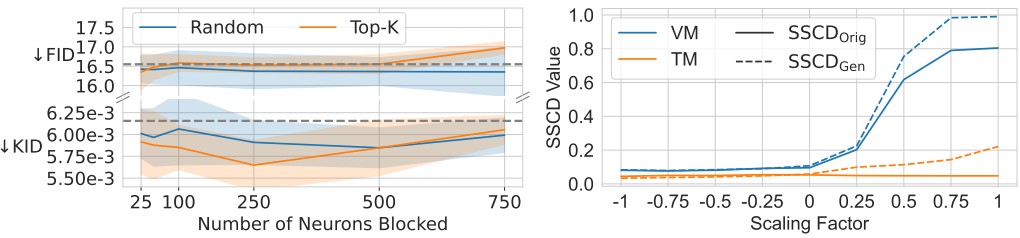

(a) Assessing image quality using FID and KID.  (b) Scaling the activations of memorization neurons.

Figure 5: **Image Quality and Sensitivity to Scaling Factor. (a)** assesses the generated images' quality when blocking an increasing number of neurons. As can be seen, the FID and KID values vary only slightly, indicating that blocking neurons identified by NEMO does not negatively affect image generation quality. Gray lines indicate the baseline without any neurons blocked. **(b)** investigates the effect of scaling the memorization neurons' activations by a scaling factor instead of deactivating them (scaling by zero). Whereas positively scaling memorization neuron activations only slightly reduces memorization, negative scaling reduces the memorization not any further.

However, approximately one-third of TM prompts are distributed across 50 or more neurons. We hypothesize that this broader distribution results from the higher variation in generated images for TM prompts, where memorization spread across multiple neurons leads to increased image diversity. In contrast, VM prompts, often memorized by a small group of neurons, consistently produce the same image without variation. More detailed plots of the identified neurons can be found in Appx. C.2.

Interestingly, we identify two neurons in the first cross-attention value layer responsible for the verbatim memorization of multiple prompts. Neuron #25 in this layer is associated with depicting people, while neuron #221 is responsible for memorizing multiple podcast covers. Together, these neurons account for memorizing 17% of our dataset's VM prompts. Similarly, neurons #507 and #517 in the third value layer are responsible for multiple TM prompts describing iPhone cases. The impact of deactivating these neurons on the image generation of memorized prompts is visualized in Appx. C.5. We also plot the distribution of the average layer-wise number of memorization neurons per prompt in Fig. 4b. Neurons responsible for VM prompts are primarily located in the value mappings of the first cross-attention layers within the U-Net's down-blocks (each block contains two cross-attention layers). A similar pattern appears for TM prompts, although value layers located deeper in the U-Net seem to play a more crucial role for TM prompts than for VM prompts.

### 4.3 Memorization Neurons Hardly Influence Non-Memorized Prompts

Until now, our focus has been on the impact of deactivating memorization neurons on memorized prompts. In this part, we investigate how these neurons influence non-memorized prompts and the overall image quality of the DM. To assess their impact, we deactivate varying numbers of memorization neurons, ordered by their frequency of occurrence as identified in our experiments, and compute the FID and KID scores on the COCO dataset. We also repeat the generations by deactivating the same number of randomly selected neurons that are not among the identified memorization neurons. As shown in Fig. 5a, there is no significant degradation in the image quality when blocking either the random neurons or the memorization neurons, even with up to 750 blocked neurons. This finding underscores the potential for pruning memorization neurons in DMs without compromising the overall image quality. The plot for the CLIP-FID metric and more detailed plots of the other two metrics, as well as an additional experiment measuring the disentanglement of the neurons in the value layers, can be found in Appx. C.3.

### 4.4 Ablation Study and Sensitivity Analysis

We further analyze the impact of each component of NEMO and its sensitivity to hyperparameter selection. We discuss the most crucial insights here, with the complete study included in Appx. C.7. First, we evaluate the impact of different memorization thresholds $\theta_{mem}$. Lowering this threshold slightly increases the number of identified neurons but has a negligible effect on performance metrics.

Selecting this threshold based on statistics computed on a holdout set provides a simple yet effective way for hyperparameter selection.

Additionally, we compare the results of using both stages of NeMo versus a setting where we only perform the initial candidate selection, skipping the refinement process. Although memorization is successfully mitigated by deactivating the initially selected neurons, the number of identified memorization neurons increases substantially, resulting in a median of 26.5 neurons (+*22.5 neurons*) for VM prompts and 674.5 neurons (+*653.5 neurons*) for TM prompts. This highlights the importance of the refinement stage in reducing the number of neurons necessary to mitigate memorization *efficiently*. To further test whether the assumptions about the statistics of memorization neurons hold, we applied NeMo to a set of 500 non-memorized prompts not used to calibrate our thresholds. As we further show, NeMo does not identify any neurons for most of the non-memorized prompts, underscoring the validity of our assumptions.

We also compared the effect of completely deactivating the identified memorization neurons to down-scaling their activations by a fixed factor. The SSCD scores in Fig. 5b, computed for different scaling factors, demonstrate that memorization is not fully mitigated when using a positive scaling factor. Conversely, negative scaling factors do not provide any additional mitigation compared to our default setting of deactivating the neurons (i.e., using a factor of zero).

## 5    Conclusion and Outlook

DMs have rapidly become a cornerstone of computer vision. Yet, problems like memorization of training samples can lead to undesired replication of potentially sensitive or copyrighted training images. Previous research has primarily focused on identifying memorized prompts and proposing mitigation strategies by adjusting the DM's input. However, there has been a lack of understanding regarding the precise location of memorization within the model.

Our research provides novel insights into the memorization mechanisms in text-to-image DMs. Unlike previous studies that focused on identifying memorized prompts, our approach, NeMo, is the first to *localize memorization* within the model and pinpoint *individual neurons* responsible for it. Traditional pruning methods [11] are orthogonal to our approach by pruning only structures to reduce the total parameter count of the model. Our memorization localization algorithm enables model providers to prune these memorization neurons, effectively mitigating memorization permanently without additional model training, which can be costly in terms of data and resources. Our mitigation can be executed without compromising the model's overall performance or the quality of the generated images, allowing model providers to deploy the resulting models without additional safeguards to prevent memorization.

There are several directions to expand and build upon our method for detecting neurons responsible for memorization. One intriguing avenue is to investigate whether an adjusted version of NeMo can detect concept neurons [23]. These neurons are not responsible for memorizing a certain prompt but for generating a particular concept. Such an approach could enable model providers and users to perform knowledge editing [15] and remove undesired concepts like violence and nudity. Another exciting application for NeMo is in large language models, also known for memorizing training samples [5]. Identifying neurons responsible for memorizing text from the training data could lead to new mitigation strategies.

Additionally, our insights could be interesting for developing new pruning algorithms for DMs to reduce the number of parameters while eliminating unintended memorization. As demonstrated in our experiments, pruning memorization neurons does not significantly impact the model's overall performance, which is crucial for effective pruning strategies.

## Acknowledgments and Disclosure of Funding

This work has been financially supported by the German Research Center for Artificial Intelligence (DFKI) project "SAINT". The project also received funding from the Initiative and Networking Fund of the Helmholtz Association in the framework of the Helmholtz AI project call under the name "PAFMIM" with funding number ZT-I-PF-5-227. Responsibility for the content of this publication lies with the author.

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

# A  Limitations

**Highly Memorized Prompts:** For certain memorized prompts, our method identifies a set of over 100 neurons. Upon closer examination, we found that in these cases, memorization is distributed across many neurons in various layers, rather than being concentrated in a small group. Even deactivating a substantial number of neurons in the network does not eliminate memorization for these prompts. However, such instances, primarily TM prompts, were rare in our experiments. We provide examples of highly memorized prompts in Appx. C.4.

**Mitigating Template Memorization Is Harder Than Verbatim Memorization:** Deactivating memorization neurons is effective for mitigating verbatim memorization, and often requires only a small number of neurons to be deactivated. In contrast, mitigating template memorization often requires deactivating more neurons, and even then, complete removal of memorization is not always possible in some rare cases. This difficulty arises because template memorization frequently results in diverse generations, with the memorized content corresponding to only certain aspects of the image. Distinguishing between these memorized parts and the remaining image parts is not always clearly achievable with our SSIM-based memorization strength. However, we emphasize that for the majority of prompts, deactivating the identified memorization neurons successfully removes template memorization.

**Runtime:** For most memorized prompts, NEMO detects the memorization neurons in a few seconds. We timed the runtime of NEMO and found that on average NEMO can identify memorization neurons for verbatim memorized prompts within $14.2$ seconds, while for template memorized prompts memorization neurons are identified in $43.7$ seconds on average. As discussed in the previous paragraph, mitigating and localizing memorization for template memorized prompts is harder. We suspect this is one reason why NEMO's runtime is slightly longer for template memorized prompts. To get further insight into how long the runtime of each part of our algorithm is, we also timed the runtime for the algorithms D1-D4 separately. The results can be seen in Tab. 2.

Table 2: NEMO can localize the neurons responsible for memorization efficiently. The average runtime (in seconds) for Alg. 1 and Alg. 2 is below one second, while the runtime for Alg. 3 is below 10 seconds. While the runtime for Alg. 4 is longer than for the other parts of NEMO, the runtime is still only 45 seconds for TM. Alg. 4 has the longest runtime for TM, since the initial candidate set is larger than for verbatim memorized samples.

| Memorization Type | A1 | A2 | A3 | A4 | Total |
|---|---|---|---|---|---|
| VM | 0.21 | 0.29 | 2.31 | 12.85 | 15.68 |
| TM | 0.18 | 0.25 | 6.09 | 39.37 | 45.90 |

# B  Experimental Details

## B.1  Hard- and Software Details

We performed all our experiments on NVIDIA DGX machines running NVIDIA DGX Server Version 5.2.0 and Ubuntu 20.04.5 LTS. The machines have 1.5 TB (machine 1) and 2 TB (machine 2) of RAM and contain NVIDIA Tesla V100 SXM3 32GB (machine 1) NVIDIA A100-SXM4-40GB (machine 2) GPUs with Intel(R) Xeon(R) Platinum 8174 (machine 1) and AMD EPYC 7742 64-core (machine 2) CPUs. We further relied on CUDA 12.1, Python 3.10.13, and PyTorch 2.2.2 with Torchvision 0.17.2 [27] for our experiments. All investigated models are publicly available on Hugging Face. For access, we used the Hugging Face diffusers library with version 0.27.1.

We provide a Dockerfile with our code to make reproducing our results easier. In addition, all training and configuration files are available to reproduce the results stated in this paper.

## B.2  Model and Dataset Details

Experiments were mainly conducted on Stable Diffusion v1-4 [32], publicly available at `https://huggingface.co/CompVis/stable-diffusion-v1-4`. All details regarding the data, training

parameters, limitations, and environmental impact are available at that URL. The model is available under the CreativeML OpenRAIL M license.

The investigated prompts originate from the LAION2B-en [35] dataset used to train the DM. The set of memorized prompts is taken from Wen et al. [46][2], who collected the prompts by using the tool of Webster [45]. The LAION dataset itself is licensed under the Creative Common CC-BY 4.0. The images of the LAION dataset might be under copyright, so we do not include them in our code base; we only provide URLs to retrieve the images directly from their source.

### B.3 Experimental Details and Hyperparameters

All images depicted throughout the paper are generated with fixed seeds, 50 inference steps, and a classifier-free guidance strength of 7 using the default DDIM scheduler. Notably, the seeds used for generating the images and computing the evaluation metrics differ from those used for our detection method NEMO to avoid seed overfitting.

During detection with NEMO, no classifier-free guidance was used, which speeds up the detection since only a single forward pass per seed is required, compared to an additional forward pass on the null-text embedding with classifier-free guidance. We always used ten different seeds for each prompt. The threshold on the SSIM memorization score was set to $\tau_{\text{mem}} = 0.428$ during the experiments in the main paper. We vary this threshold and analyze its impact in our sensitivity analysis in Appx. C.7.

We run all experiments – detection with NEMO and the generations for the metric computations – with half-precision (float16) to reduce the memory consumption and speed up the computations.

### B.4 Structural Similarity Index Measure (SSIM)

We quantify the memorization strength during our experiments using the structural similarity index measure commonly used in the computer vision domain to assess the similarity between image pairs. Our memorization score is computed as follows: Let $x_T \sim N(\mathbf{0}, \mathbf{I})$ be the initial noisy image. Let $\epsilon_\theta(x_T, T, y)$ further denote the initial noise prediction without any scaling by the scheduler. We found that the normalized difference between the initial noise and the first noise prediction $\delta = \epsilon_\theta(x_T, T, y) - x_T$ for memorized prompts is substantially more consistent for different seeds than for non-memorized prompts. To detect the grade of memorization, we, therefore, use the similarity between the noise differences $\delta^{(i)}$ and $\delta^{(j)}$ generated with seeds $i$ and $j$ as a proxy. We measure the similarity with the common structural similarity index measure (SSIM) [44]. The SSIM $\in [0, 1]$ between two noise differences $\delta^{(i)}$ and $\delta^{(j)}$ is defined by

$$\text{SSIM}(\delta^{(i)}, \delta^{(j)}) = \frac{(2\mu_i\mu_j + C_1)(2\sigma_{ij} + C_2)}{(\mu_i^2 + \mu_j^2 + C_1)(\sigma_i^2 + \sigma_y^2 + C_2)}. \tag{4}$$

The parameters $\mu_i$, $\mu_j$ and $\sigma_i^2$, $\sigma_j^2$ denote the mean and variance of the pixels in $\delta^{(i)}$ an $\delta^{(j)}$, respectively. Likewise, $\sigma_{ij}$ denotes the covariance between the images. Following the original paper, $C_1$ and $C_2$ are small constants added for numerical stability.

A higher SSIM indicates higher similarity between the noise differences, reflecting a higher degree of memorization. Notably, the SSIM computation only requires a single denoising step per seed, which makes the process fast.

Indeed, the SSIM score itself can be used to detect memorization. We visualized the different SSIM score distributions for non-memorized and memorized prompts in Fig. 2a. To underline this observation with quantitative metrics, we ran additional experiments to explore the detection capabilities of our SSIM-based method. We measured the efficiency of the SSIM score for memorization detection on our dataset of memorized and non-memorized prompts. Without extensive hyperparameter tuning, this detection method achieves an AUROC of $98\%$, an accuracy of $94.2\%$ (using a naive threshold of 0.5 on the SSIM similarity), and a TPR@1%FPR of $87.6\%$. Since the amount of memorization of template memorized prompts varies significantly, we repeated the computation for detecting the verbatim memorized prompts. Here, the SSIM approach even achieves an AUROC of $99.64\%$, an accuracy of $97.39\%$ (with a threshold of 0.5), and a TPR@1%FPR of $98.21\%$. These results indicate that the SSIM score can also be used to detect memorization reliably in the first place. However, detection is not the focus of the paper but the localization on the neuron level.

---

[2]Available at `https://github.com/YuxinWenRick/diffusion_memorization`.

# C  Additional Results

## C.1  Distinguishing Between Different Types of Memorization

Webster [45] distinguished between *verbatim* and *template memorized* prompts. Verbatim memorized prompts lead to the exact reconstruction of training samples, while template-memorized prompts replicate the composition and structure of the training image. To provide a more fine-grained analysis of our results, we classify the prompts in our dataset into these two categories. We distinguish between both types by computing the SSCD [28] scores between the original training image and ten generations with different seeds. We then classify a prompt as *verbatim memorized* if the maximum SSCD score computed as cosine similarity exceeds a threshold of $0.7$ and as *template memorized* otherwise. Fig. 6 plots the distribution of SSCD scores for both datasets. We manually inspected and classified the prompts where the original training image is no longer available (16 out of 500).

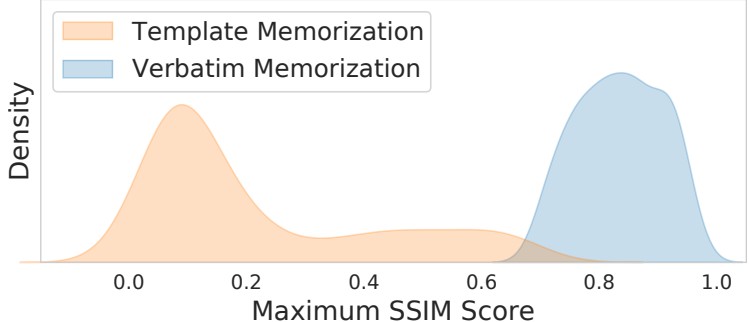

Figure 6: We compare the maximum SSCD score between ten generated images and the original training sample. We categorize the memorized prompts into verbatim memorized if the SSCD score exceeds $0.7$ and into template memorized prompts otherwise.

## C.2  Detailed Analysis of the Distribution of Memorization Neurons

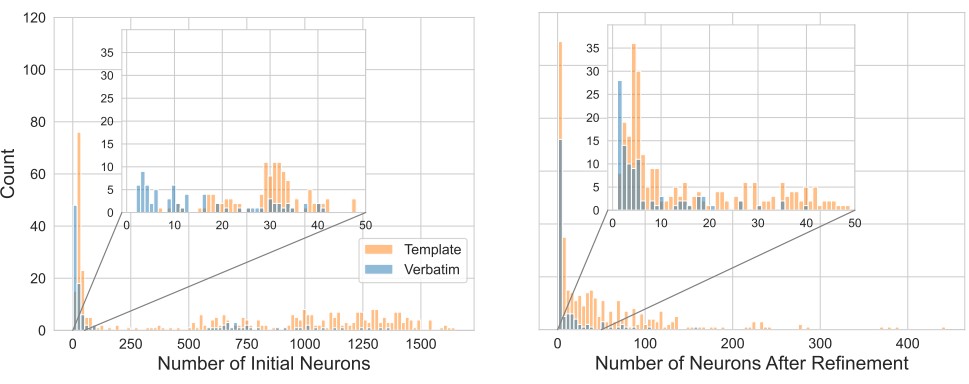

(a) Number of initial neurons per prompt.

(b) Number of neurons per prompt after refinement.

Figure 7: **Distribution of Memorization Neurons**. We show the number of prompts that are memorized by a fixed number of neurons. **(a)** plots the number of neurons found in the initial neuron selection. **(b)** shows the number of neurons after refinement. As we can observe, the refinement step drastically reduces the number of found memorization neurons for both the template and the verbatim memorized prompts.

## C.3 Detailed Quality Analysis

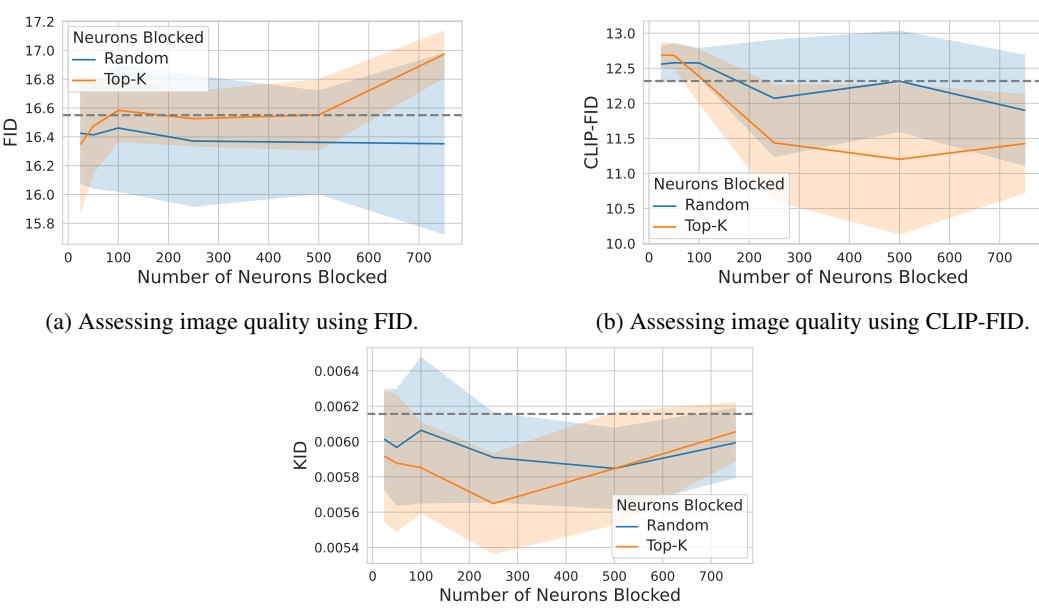

(a) Assessing image quality using FID.

(b) Assessing image quality using CLIP-FID.

(c) Assessing image quality using KID.

Figure 8: **Image Quality Does Not Degrade When Deactivating Memorization Neurons**. Depicted are the generated images' FID, CLIP-FID (FID calculated using a CLIP model), and KID scores when blocking an increasing number of neurons. For all three metrics, smaller values are better. As can be seen, the FID, KID, and CLIP-FID values vary only slightly, indicating that blocking neurons identified by NEMO does not negatively affect image generation quality. Gray lines indicate baselines without any neurons blocked. We repeated the experiment with five different seeds. Depicted are the mean values and the standard deviation.

Table 3: We measured the disentanglement of neurons for deactivating top-k memorization neurons or randomly selected neurons. More specifically, we collected and compared the attention layer outputs for 500 non-memorized prompts with and without neurons deactivated and 3 seeds. We then computed the average cosine similarity between corresponding outputs. The high similarities show that blocking neurons only has a negligible impact on the outputs of the attention layer, suggesting that other neurons can substitute the functionality of blocked neurons on non-memorized prompts.

| Num. Blocked Neurons | 1 | | 5 | | 10 | | 25 | | 50 | | 100 | | 150 | |
|---|---|---|---|---|---|---|---|---|---|---|---|---|---|---|
| Block Neuron Type | Random | Top-K | Random | Top-K | Random | Top-K | Random | Top-K | Random | Top-K | Random | Top-K | Random | Top-K |
| 1 | 1.0 | 1.0 | 1.0 | 0.99 | 1.0 | 0.98 | 0.99 | 0.98 | 0.99 | 0.98 | 0.99 | 0.98 | 0.99 | 0.98 |
| 2 | 1.0 | 0.99 | 1.0 | 0.99 | 1.0 | 0.99 | 0.99 | 0.98 | 0.99 | 0.98 | 0.99 | 0.98 | 0.99 | 0.98 |
| 3 | 1.0 | 1.0 | 0.99 | 0.99 | 0.99 | 0.99 | 0.99 | 0.99 | 0.99 | 0.99 | 0.99 | 0.99 | 0.99 | 0.99 |
| 4 | 1.0 | 1.0 | 1.0 | 0.99 | 0.99 | 0.99 | 0.99 | 0.99 | 0.99 | 0.99 | 0.99 | 0.99 | 0.99 | 0.99 |
| 5 | 1.0 | 1.0 | 1.0 | 0.99 | 0.99 | 0.99 | 0.99 | 0.99 | 0.99 | 0.99 | 0.99 | 0.99 | 0.99 | 0.99 |
| 6 | 1.0 | 0.99 | 0.99 | 0.99 | 0.99 | 0.99 | 0.99 | 0.99 | 0.99 | 0.99 | 0.99 | 0.99 | 0.99 | 0.99 |
| 7 | 0.99 | 0.99 | 0.99 | 0.99 | 0.99 | 0.99 | 0.99 | 0.99 | 0.99 | 0.99 | 0.99 | 0.99 | 0.99 | 0.99 |

We ran additional experiments to analyze the impact and effects of deactivating neurons in the value layers. The result can be seen in Tab. 3. We deactivated a varying number of neurons, either randomly selected or the top memorization neurons, and measured the similarity between the outputs of the cross-attention blocks with deactivated neurons and all neurons activated. Our results, stated in the rebuttal PDF, show that even deactivating a large number of neurons only impacts the attention outputs marginally. We, therefore, conclude that other neurons can replace the functionality of specific neurons on non-memorized prompts and that neurons in the value layers act rather independently. Yet, for neurons memorizing specific prompts, this memorizing functionality is not replaced by other neurons, explaining the mitigation effect of deactivating these neurons.

## C.4 Highly Memorized Prompts

**No Blocked Neurons**  **Blocked Memorization Neurons**

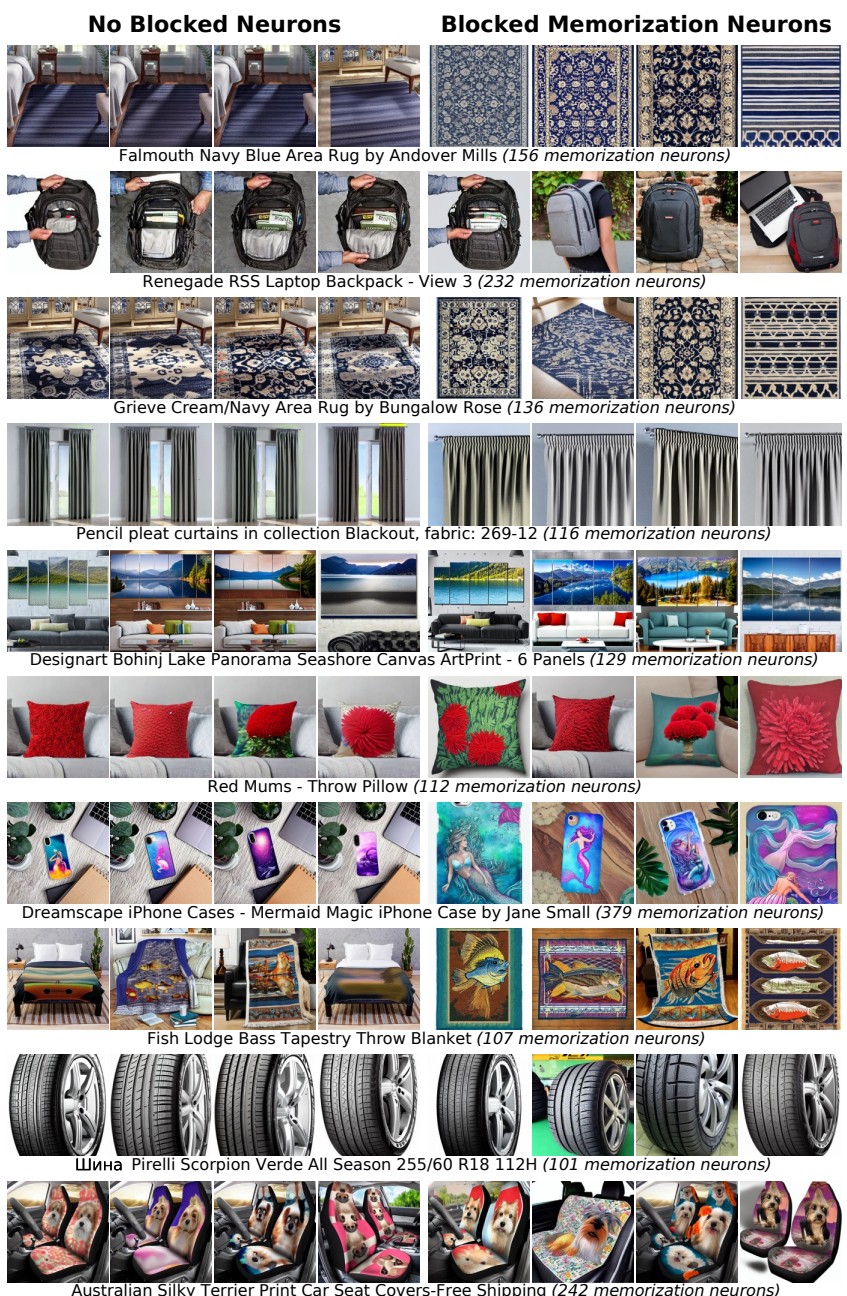

Figure 9: Memorization of highly memorized prompts is distributed across many neurons in various layers, rather than concentrated in a small group of neurons. We show examples of such prompts and the impact of deactivating the identified memorization neurons. The number of memorization neurons in each case is stated behind each prompt.

## C.5 Examples for Memorization of Single Neurons

To illustrate that some single neurons are responsible for memorizing multiple training prompts, we generated images with and without these specific neurons deactivated. In Fig. 10 and Fig. 11, we only deactivate a single neuron each in the first value layer, whereas in Fig. 12, we deactivate two neurons in the third value layer.

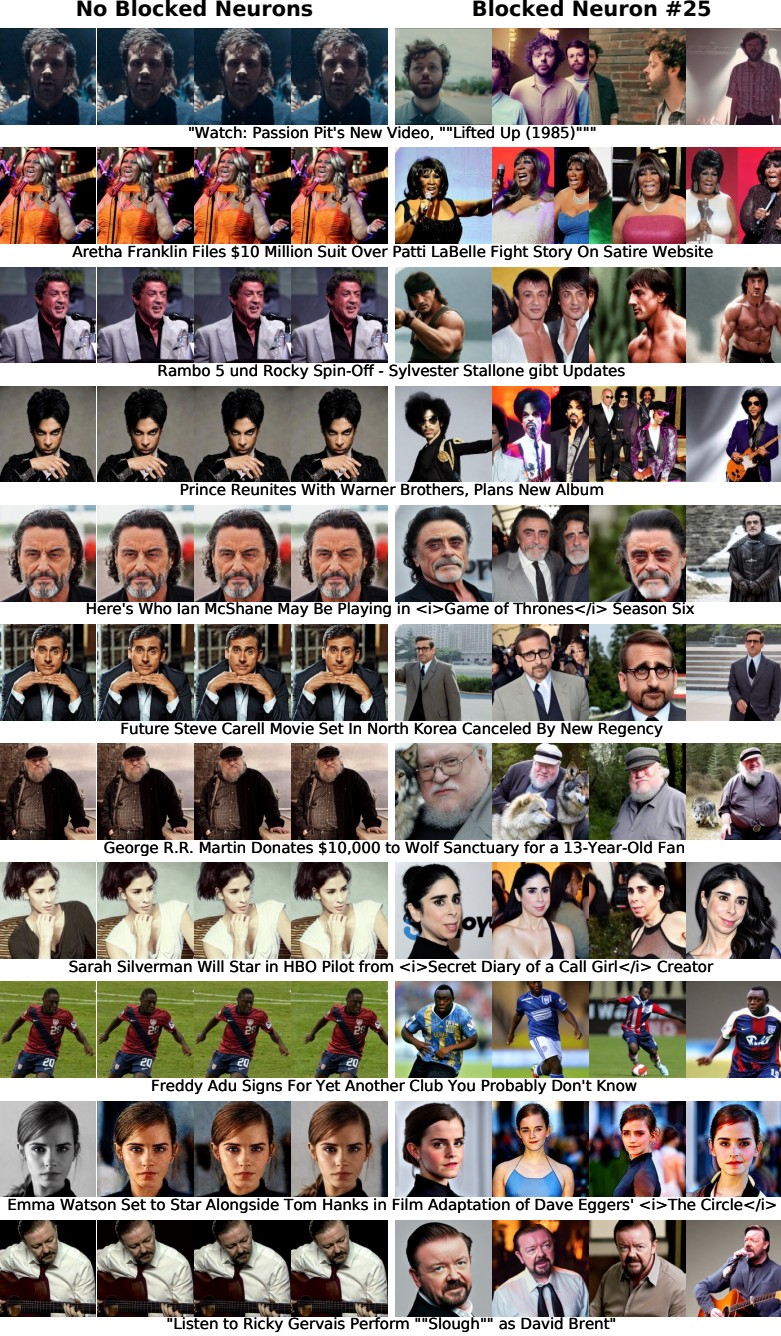

Figure 10: We found that **neuron #25 in the first cross-attention layer's value mapping** is responsible for verbatim memorization of multiple prompts, all associated with depicting people. Deactivating this single neuron mitigates the memorization and introduces diversity into the images (right columns) compared to images generated with all neurons active (left columns). Generations were conducted with seeds different from the seeds used for the neuron localization process.

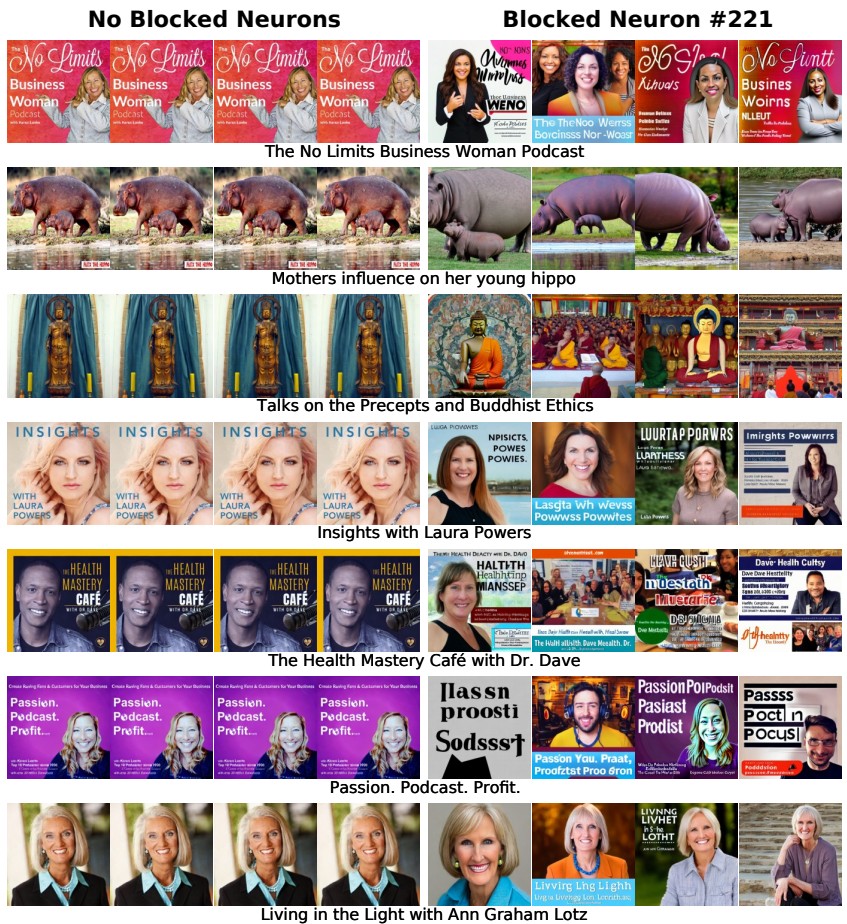

Figure 11: We found that **neuron #25 in the first cross-attention layer's value mapping** is responsible for verbatim memorization of multiple prompts, all associated with depicting people. Deactivating this single neuron mitigates the memorization and introduces diversity into the images (right columns) compared to images generated with all neurons active (left columns). Generations were conducted with seeds different from the seeds used for the neuron localization process.

**No Blocked Neurons**    **Blocked Neurons #507 & #517**

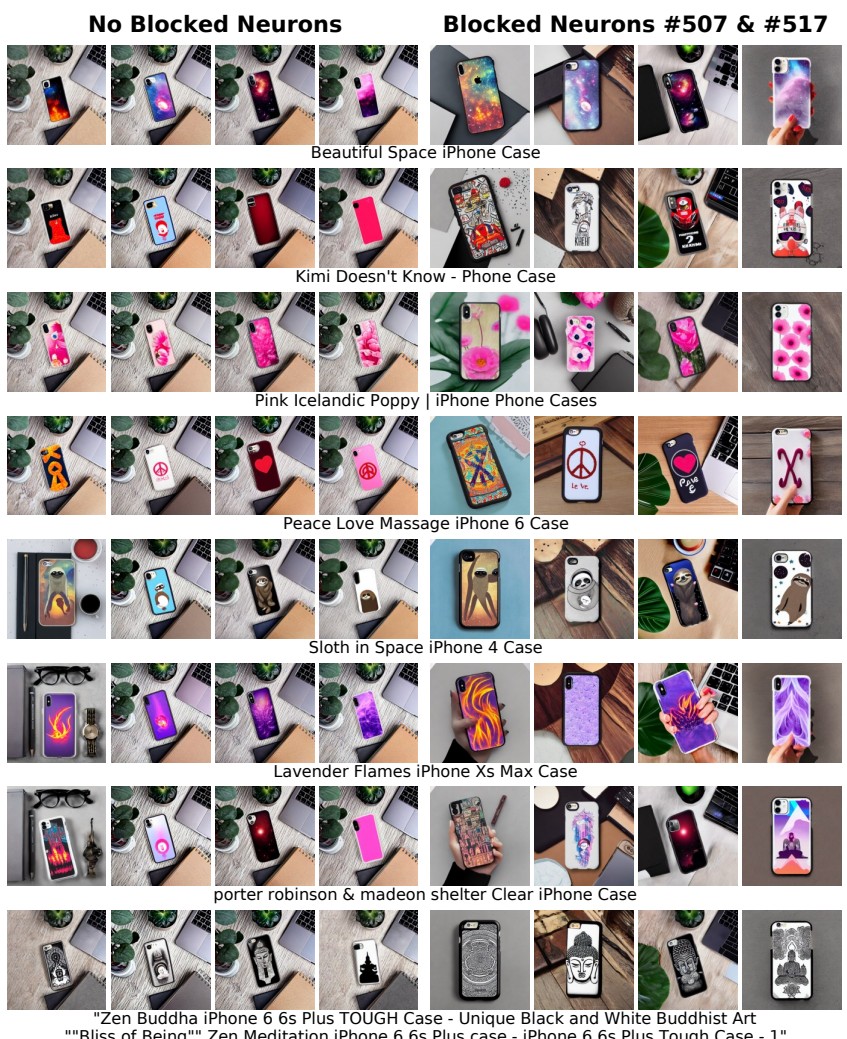

Figure 12: We found that **neurons #507 and #517 in the third cross-attention layer's value mapping** is responsible for template memorization of multiple prompts describing iPhone cases. Deactivating these two neurons mitigates the template memorization and introduces diversity into the images (right columns) compared to images generated with all neurons active (left columns). Image generations were conducted with a fixed seed different from the seed used for the neuron localization process.

## C.6 Comparison of Initial Noise Predictions

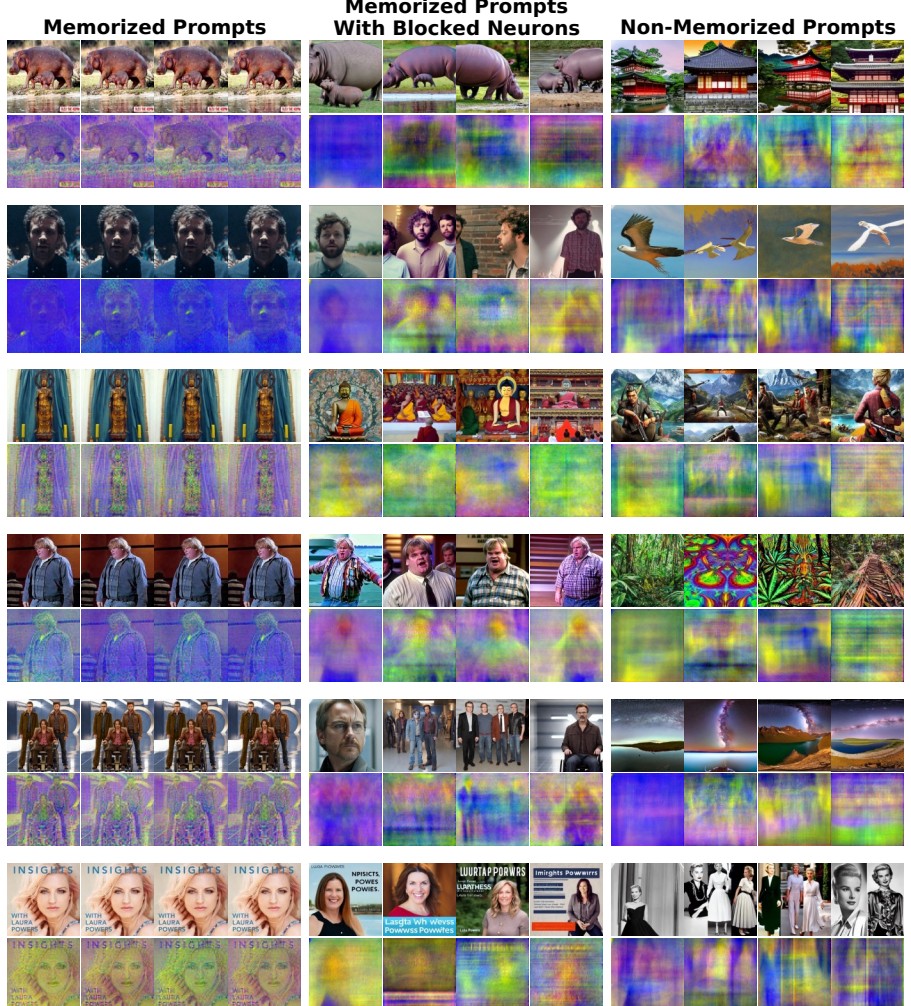

Figure 13: Visualizations for generated images and the **noise differences between the predicted noise after the first denoising step and the initial Gaussian noise**. Noise differences for memorized prompts (left column) have low diversity and are already structurally similar to the final image. The noise differences for non-memorized prompts (right column) show no clear structure and differ substantially for different noise initializations. Deactivating the memorization neurons detected with NEMO (middle column) removes the structure in the initial noise differences and adds more diversity, leading to diverse image generations.

## C.7 Ablation Study and Sensitivity Analysis

We conduct an ablation study to investigate the impact of the individual components of NEMO. Additionally, we analyze the sensitivity of the memorization threshold $\tau_{\text{mem}}$ and explore alternatives to deactivating neurons by setting their activations to zero. The results for the various settings are presented in Tab. 4.

The first two rows provide evaluation results for the model with *all neurons active* and with *randomly deactivated neurons*. Both scenarios exhibit strong memorization. The third row shows the results of blocking the neurons identified as memorizing by NEMO, using the threshold $\tau_{\text{mem}} = 0.428$ as specified in the main paper. This threshold corresponds to the mean SSIM memorization plus one standard deviation, calculated on a holdout set of 50,000 non-memorized LAION prompts. In row four, we repeat this setting using classifier-free guidance (*CFG*) with a guidance strength of 7.0, as opposed to our default setting without CFG. Detection with CFG further reduces the number of detected memorization neurons. However, the SSCD scores indicate slightly increased memorization after deactivating the identified neurons. Additionally, running NEMO with CFG doubles the number of forward passes in the U-Net since a separate noise prediction is generated for each initial seed.

Rows five to seven display the results of *varying the memorization threshold $\tau_{\text{mem}}$*. Specifically, we adjust the threshold to one standard deviation below the mean SSIM score, to the mean, and two standard deviations above the mean. A lower threshold identifies more neurons. However, for lower thresholds, the metrics are comparable to those obtained with our default threshold value ($\tau_{\text{mem}} = 0.428$). Increasing the threshold reduces the number of identified neurons but slightly increases memorization, as measured by the SSCD scores. Thus, a trade-off exists between reducing the number of identified memorization neurons and their memorization mitigation effect. In addition, we provide heat maps to directly compare the impact of different thresholds $\tau_{\text{mem}}$ used during the initial selection and the refinement step in Fig. 14. Fig. 15 further compares the SSCD scores for varying the threshold values.

Rows eight and nine examine the impact of *removing the refinement step* or incorporating *no neurons with top-k activations* during the initial selection. As anticipated, without refinement, the number of identified neurons increases substantially. Despite this, the various metrics remain comparable to those obtained after the refinement step, even with more neurons deactivated. This underscores the robustness of image generations against pruning out-of-distribution (OOD) neurons. Without the top-k selection, NEMO identifies a larger set of neurons. However, deactivating these neurons does not mitigate memorization as effectively as with a top-k search. Notably, for template verbatim prompts, the $\text{SSCD}_{\text{Gen}}$ is substantially higher without top-k, indicating increased memorization.

In the remaining rows, we explore the impact of *scaling the activations of memorization neurons* instead of deactivating them. With negative scaling factors, the results are comparable to those of completely deactivating the neurons. For positive scaling factors, however, the generated images demonstrate higher degrees of memorization, with a scaling factor of 0.75 having almost no influence on memorization.

We also apply NEMO to a set of 500 LAION *non-memorized prompts*, different from the 50,000 prompts used to set the memorization threshold. For 442 of these prompts, NEMO identified no memorization neurons, which is to be expected since these prompts show no memorization behavior. For the remaining prompts, a median of $62 \pm 27$ neurons was found.

Table 4: **Quantitative Results of Our Ablation Study and Sensitivity Analysis.**

| Setting | Memorization Type | Deactivated Neurons | $\downarrow SSCD_{\text{Orig}}$ | $\downarrow SSCD_{\text{Gen}}$ | $\downarrow D_{\text{SSCD}}$ | $\uparrow A_{\text{CLIP}}$ |
|---|---|---|---|---|---|---|
| All Neurons Activate (Default) | Verbatim | 0 | $0.83 \pm 0.16$ | $1.0 \pm 0.0$ | $0.99 \pm 0.01$ | $0.32 \pm 0.02$ |
| | Template | 0 | $0.04 \pm 0.04$ | $1.0 \pm 0.0$ | $0.17 \pm 0.06$ | $0.31 \pm 0.02$ |
| Deactivating Random Neurons | Verbatim | $4 \pm 3$ | $0.80 \pm 0.11$ | $0.999 \pm 0.0$ | $0.99 \pm 0.01$ | $0.32 \pm 0.02$ |
| | Template | $21 \pm 18$ | $0.05 \pm 0.04$ | $0.997 \pm 0.0$ | $0.16 \pm 0.06$ | $0.31 \pm 0.02$ |
| Default Values ($\tau_{\text{mem}} = 0.428$) | Verbatim | $4 \pm 3$ | $0.09 \pm 0.06$ | $0.10 \pm 0.07$ | $0.16 \pm 0.06$ | $0.31 \pm 0.02$ |
| | Template | $21 \pm 18$ | $0.05 \pm 0.03$ | $0.05 \pm 0.04$ | $0.12 \pm 0.05$ | $0.31 \pm 0.02$ |
| With Classifier-Free Guidance | Verbatim | $3 \pm 2$ | $0.09 \pm 0.06$ | $0.14 \pm 0.07$ | $0.15 \pm 0.06$ | $0.31 \pm 0.02$ |
| | Template | $6 \pm 5$ | $0.05 \pm 0.03$ | $0.12 \pm 0.07$ | $0.11 \pm 0.04$ | $0.32 \pm 0.02$ |
| $\tau_{\text{mem}} = \mu - 1\sigma = 0.288$ | Verbatim | $10.5 \pm 9.5$ | $0.08 \pm 0.06$ | $0.14 \pm 0.06$ | $0.15 \pm 0.05$ | $0.32 \pm 0.02$ |
| | Template | $32.0 \pm 27$ | $0.06 \pm 0.03$ | $0.10 \pm 0.07$ | $0.14 \pm 0.05$ | $0.31 \pm 0.03$ |
| $\tau_{\text{mem}} = \mu = 0.358$ | Verbatim | $6 \pm 5$ | $0.10 \pm 0.06$ | $0.14 \pm 0.07$ | $0.16 \pm 0.05$ | $0.31 \pm 0.02$ |
| | Template | $30 \pm 25$ | $0.06 \pm 0.03$ | $0.11 \pm 0.07$ | $0.13 \pm 0.04$ | $0.31 \pm 0.02$ |
| $\tau_{\text{mem}} = \mu + 2\sigma = 0.498$ | Verbatim | $3 \pm 2$ | $0.10 \pm 0.06$ | $0.15 \pm 0.07$ | $0.15 \pm 0.06$ | $0.32 \pm 0.02$ |
| | Template | $7 \pm 6$ | $0.06 \pm 0.03$ | $0.12 \pm 0.04$ | $0.12 \pm 0.04$ | $0.31 \pm 0.03$ |
| No Refinement | Verbatim | $26.5 \pm 22.5$ | $0.07 \pm 0.05$ | $0.11 \pm 0.06$ | $0.15 \pm 0.06$ | $0.32 \pm 0.02$ |
| | Template | $674.5 \pm 624.5$ | $0.04 \pm 0.03$ | $0.09 \pm 0.05$ | $0.13 \pm 0.04$ | $0.31 \pm 0.02$ |
| No top-k Selection | Verbatim | $11 \pm 10$ | $0.11 \pm 0.05$ | $0.21 \pm 0.13$ | $0.16 \pm 0.04$ | $0.32 \pm 0.02$ |
| | Template | $30 \pm 23$ | $0.05 \pm 0.03$ | $0.41 \pm 0.32$ | $0.13 \pm 0.03$ | $0.31 \pm 0.02$ |
| Scaling Factor $0.75$ | Verbatim | $4 \pm 3$ | $0.79 \pm 0.12$ | $0.995 \pm 0.00$ | $0.96 \pm 0.04$ | $0.32 \pm 0.02$ |
| | Template | $21 \pm 18$ | $0.05 \pm 0.04$ | $0.966 \pm 0.03$ | $0.15 \pm 0.05$ | $0.31 \pm 0.02$ |
| Scaling Factor $0.5$ | Verbatim | $4 \pm 3$ | $0.62 \pm 0.29$ | $0.97 \pm 0.02$ | $0.67 \pm 0.33$ | $0.32 \pm 0.01$ |
| | Template | $21 \pm 18$ | $0.05 \pm 0.03$ | $0.83 \pm 0.15$ | $0.14 \pm 0.04$ | $0.31 \pm 0.02$ |
| Scaling Factor $0.25$ | Verbatim | $4 \pm 3$ | $0.20 \pm 0.17$ | $0.32 \pm 0.25$ | $0.21 \pm 0.12$ | $0.32 \pm 0.02$ |
| | Template | $21 \pm 18$ | $0.05 \pm 0.03$ | $0.23 \pm 0.16$ | $0.12 \pm 0.04$ | $0.32 \pm 0.02$ |
| Scaling Factor $-0.25$ | Verbatim | $4 \pm 3$ | $0.09 \pm 0.06$ | $0.12 \pm 0.06$ | $0.15 \pm 0.05$ | $0.32 \pm 0.02$ |
| | Template | $21 \pm 18$ | $0.05 \pm 0.03$ | $0.09 \pm 0.06$ | $0.13 \pm 0.04$ | $0.31 \pm 0.02$ |
| Scaling Factor $-0.5$ | Verbatim | $4 \pm 3$ | $0.08 \pm 0.05$ | $0.12 \pm 0.06$ | $0.17 \pm 0.06$ | $0.31 \pm 0.02$ |
| | Template | $21 \pm 18$ | $0.05 \pm 0.03$ | $0.08 \pm 0.05$ | $0.13 \pm 0.04$ | $0.31 \pm 0.02$ |
| Scaling Factor $-0.75$ | Verbatim | $4 \pm 3$ | $0.08 \pm 0.05$ | $0.11 \pm 0.06$ | $0.17 \pm 0.06$ | $0.31 \pm 0.02$ |
| | Template | $21 \pm 18$ | $0.05 \pm 0.03$ | $0.08 \pm 0.05$ | $0.14 \pm 0.05$ | $0.31 \pm 0.02$ |
| Scaling Factor $-1$ | Verbatim | $4 \pm 3$ | $0.08 \pm 0.05$ | $0.11 \pm 0.07$ | $0.16 \pm 0.06$ | $0.31 \pm 0.02$ |
| | Template | $21 \pm 18$ | $0.04 \pm 0.03$ | $0.07 \pm 0.05$ | $0.14 \pm 0.05$ | $0.30 \pm 0.02$ |

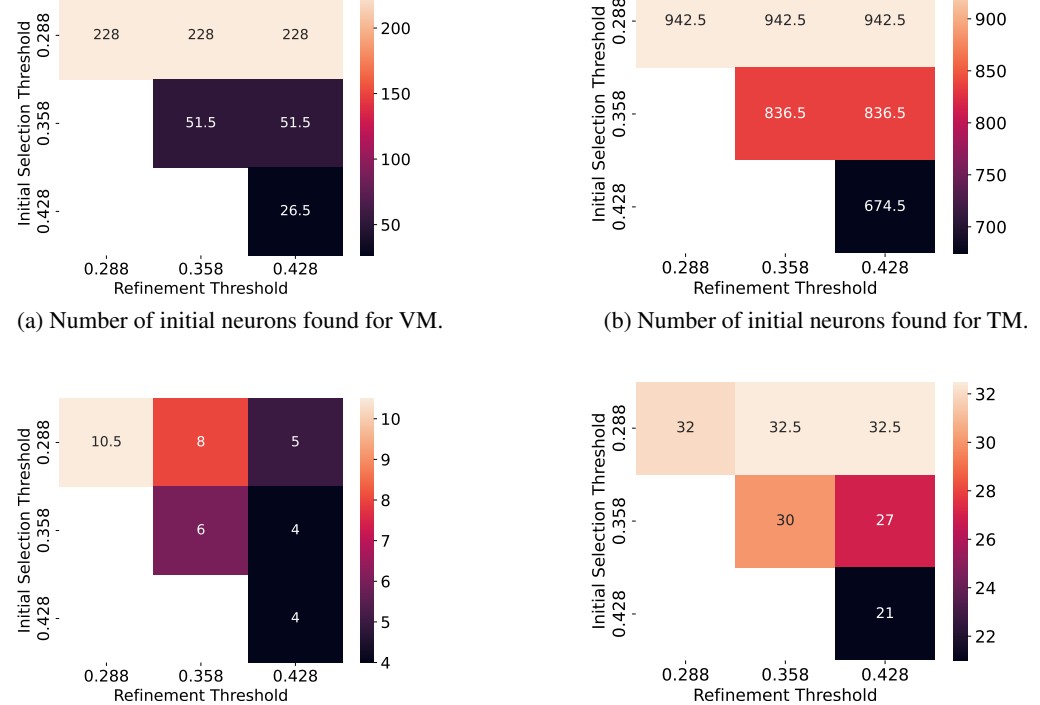

(a) Number of initial neurons found for VM.

(b) Number of initial neurons found for TM.

(c) Number of refined neurons found for VM.

(d) Number of refined neurons found for TM.

Figure 14: **Number of neurons found with different initial and refinement thresholds** $\tau_{\mathbf{mem}}$. The left plots show the results for verbatim memorization prompts, while the right plots show the results for template memorization prompts. The refinement step significantly reduces the number of identified neurons across all threshold combinations. Notably, using $0.428$ for both the initial selection and refinement thresholds results in the smallest set of identified neurons.

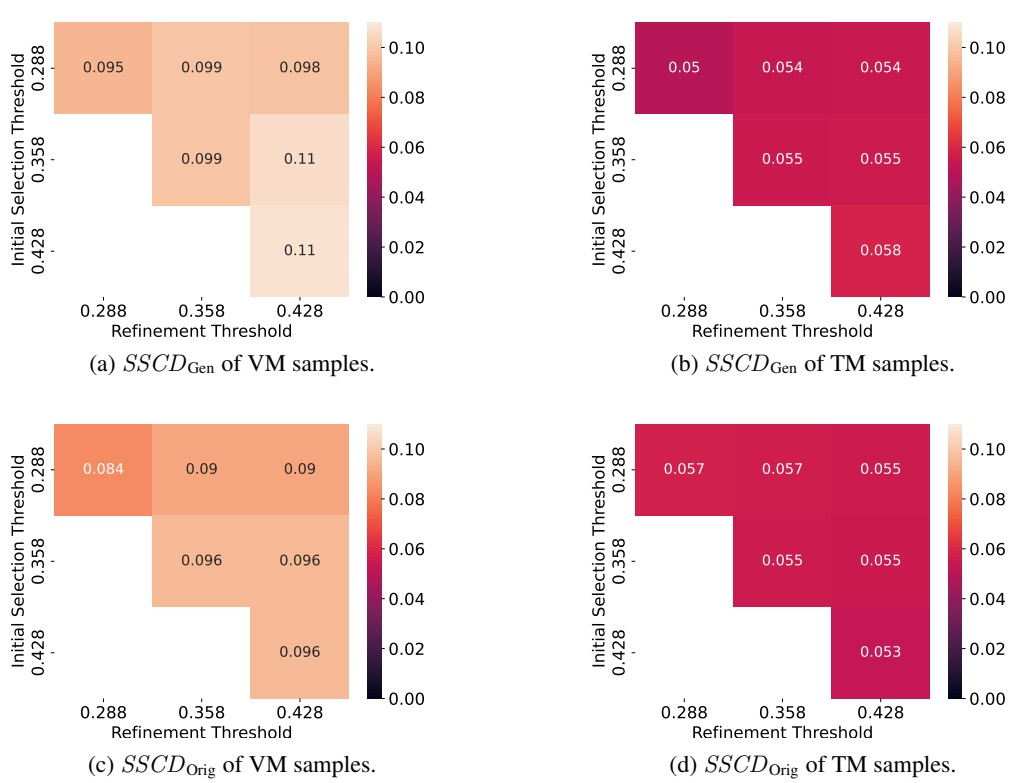

(a) $SSCD_{\text{Gen}}$ of VM samples.

(b) $SSCD_{\text{Gen}}$ of TM samples.

(c) $SSCD_{\text{Orig}}$ of VM samples.

(d) $SSCD_{\text{Orig}}$ of TM samples.

Figure 15: **SSCD memorization scores with different initial and refinement thresholds $\tau_{\text{mem}}$.** The left plots show the results for verbatim memorization prompts, while the right plots show the results for template memorization prompts. The value of the thresholds does not seem to have a high impact on the memorization scores. Since higher thresholds identify much less memorization neurons, choosing a threshold of $\tau_{\text{mem}} = 0.428$ is a valid choice.

## C.8 Ablation of Individual Key and Value Layers

During our experiments in the main paper, we limit our search with NEMO for memorization neurons to cross-attention value layers in the down- and mid-blocks of the U-Net. To motivate this decision, we perform an analysis of the influence of neurons in the individual key and value layers of different cross-attention blocks. Let us first recall the computation performed in attention layers:

$$\text{Attention}(Q, K, V) = \text{softmax}\left(\frac{QK^T}{\sqrt{d}}\right) \cdot V. \tag{5}$$

The computed key and query matrices $K$ and $Q$ are used to calculate the attention scores, i.e., the weighting of the components in the value matrix $V$. In the cross-attention layers, the query matrix $Q$ is computed by linearly mapping the current feature maps from the previous U-Net layer. Therefore, information from the textual guidance is only indirectly contained, i.e., of earlier layers or the U-Nets input feature map after the first denoising step. Therefore, we can exclude neurons in the query mapping layers since we aim to identify neurons directly responsible for memorization. The neurons in the key mapping layers directly process the text embeddings to compute the attention scores. However, strong interdependencies exist between the activations of different neurons through the nature of the softmax function. The impact each neuron's activation has on the computed attention score also depends on the activations of all other neurons from the same layer. Removing a single neuron, i.e., setting its activation to zero, does not necessarily imply substantial changes in the attention scores and the corresponding weighting of features from the value mapping layer.

The value mapping layers, however, also directly process the text embeddings, but there is no direct interdependence between the activations of different neurons. Consequently, setting the activations of individual neurons in value layers to zero directly blocks the information flow from the text embeddings. We hypothesize that the neurons in the value layers are mainly responsible for memorizing the text embeddings of specific prompts.

We evaluate this assumption by taking a set of 100 memorized prompts, generating ten samples for each prompt, and comparing the impact of removing neurons from different layers. More specifically, we remove all activations of individual key and value mapping layers, i.e., setting the output vectors of these layers to zero while keeping all other parts of the model untouched. We then compare the generated images with removed activations to the original training images. Fig. 16 plots the resulting SSCD similarity scores for deactivating individual value (top row) and key (bottom row) layers. We distinguish between verbatim (left column) and template (right column) prompts. The plots show the maximum and median SSCD scores and the deviations for the median scores. We decided not to plot deviations for the maximum score to improve readability. However, deviations are comparable to the median scores. Baselines computed without any deactivated neurons are plotted as dashed lines.

Stable Diffusion contains six cross-attention layers in the down-blocks, one in the mid-block, and nine in the up-blocks. The vertical lines indicate the separation between the different blocks. For the value layers, the layers with indexes 1 (down-lock) and 7 (mid-block) have the highest impact, whereas layers later in the network hardly change the SSCD scores. Also, the effect of the remaining layers in the down-blocks is small on their own. However, we expect there to be entwined effects between deactivating neurons in different layers, which is why we also searched for memorization neurons in these down-block layers.

For the key layers, particularly layers 4 and 6 in the down-blocks have the strongest impact on the generated images. However, removing these layers often produces images that no longer align with the concepts in the prompt or degrades the image quality, both leading to lower SSCD scores. We quantify this behavior by computing the alignments between the generated images and the corresponding input prompts in Fig. 17. While deactivating individual value layers only slightly decreases the alignment scores, deactivating some key layers substantially reduces the alignment. To further illustrate this fact, we plot some of the generated images for deactivating individual value layers in Fig. 18 and for key layers in Fig. 19.

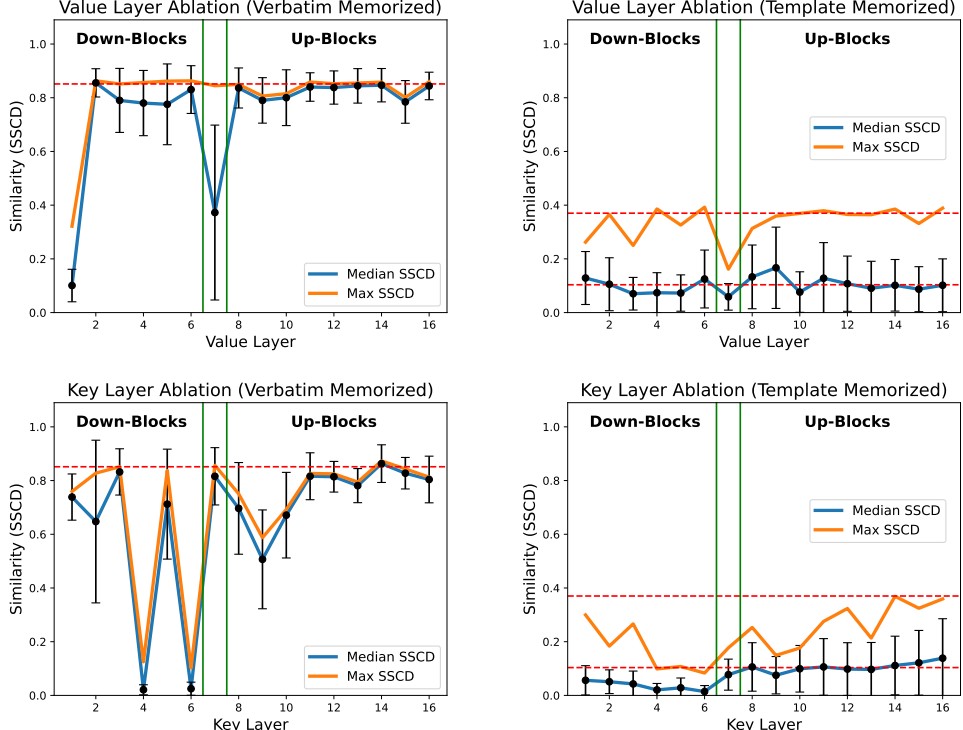

Figure 16: **SSCD similarity scores between memorized generations and the corresponding training samples.** Scores are computed for 100 prompts and ten different seeds per generation. We then take the maximum and median scores of each prompt. During the generation, we deactivated individual value and key layers of the cross-attention blocks in the network. A lower SSCD score indicates a lower similarity between generated and training images. Dashed lines denote the median and the maximum SSCD baselines for images generated without deactivating any neurons. For verbatim memorized prompts, both baselines are close, which is why we only plot the median SSCD baseline.

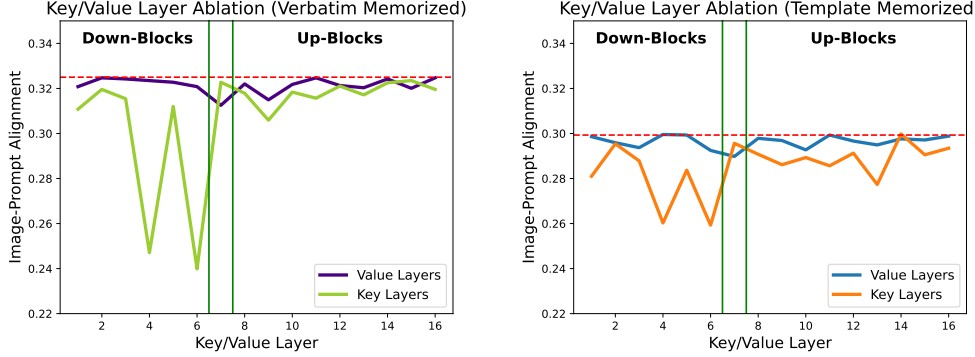

Figure 17: **CLIP alignment scores between memorized generations and the corresponding input prompt.** Scores are computed for 100 prompts and ten different seeds per generation. We then take the median alignment scores of each prompt. During the generation, we deactivated individual value and key layers of the cross-attention blocks in the network. A higher alignment score indicates a better representation of the prompt concepts in the generated images. Dashed lines denote the median alignment scores for images generated without deactivating any neurons. For both types of memorization, deactivating value layers decreases the alignment only slightly, whereas deactivating some key layers substantially reduces the alignment.

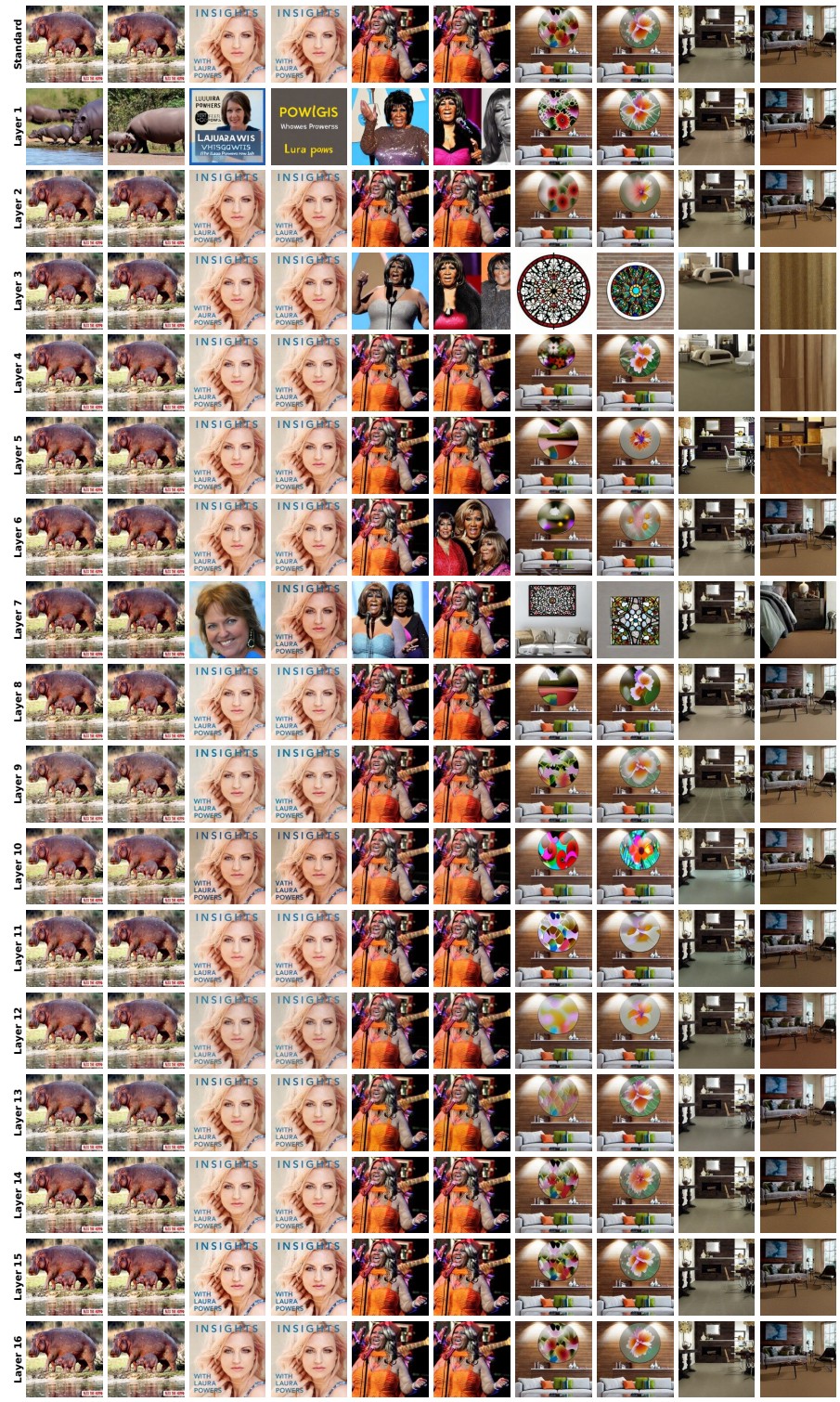

Figure 18: **Images generated with memorized prompts with deactivated individual value layers.**
Whereas the *standard* row shows generations with keeping all neurons active, the following rows
depict results for deactivating all neurons in a specific value layer.

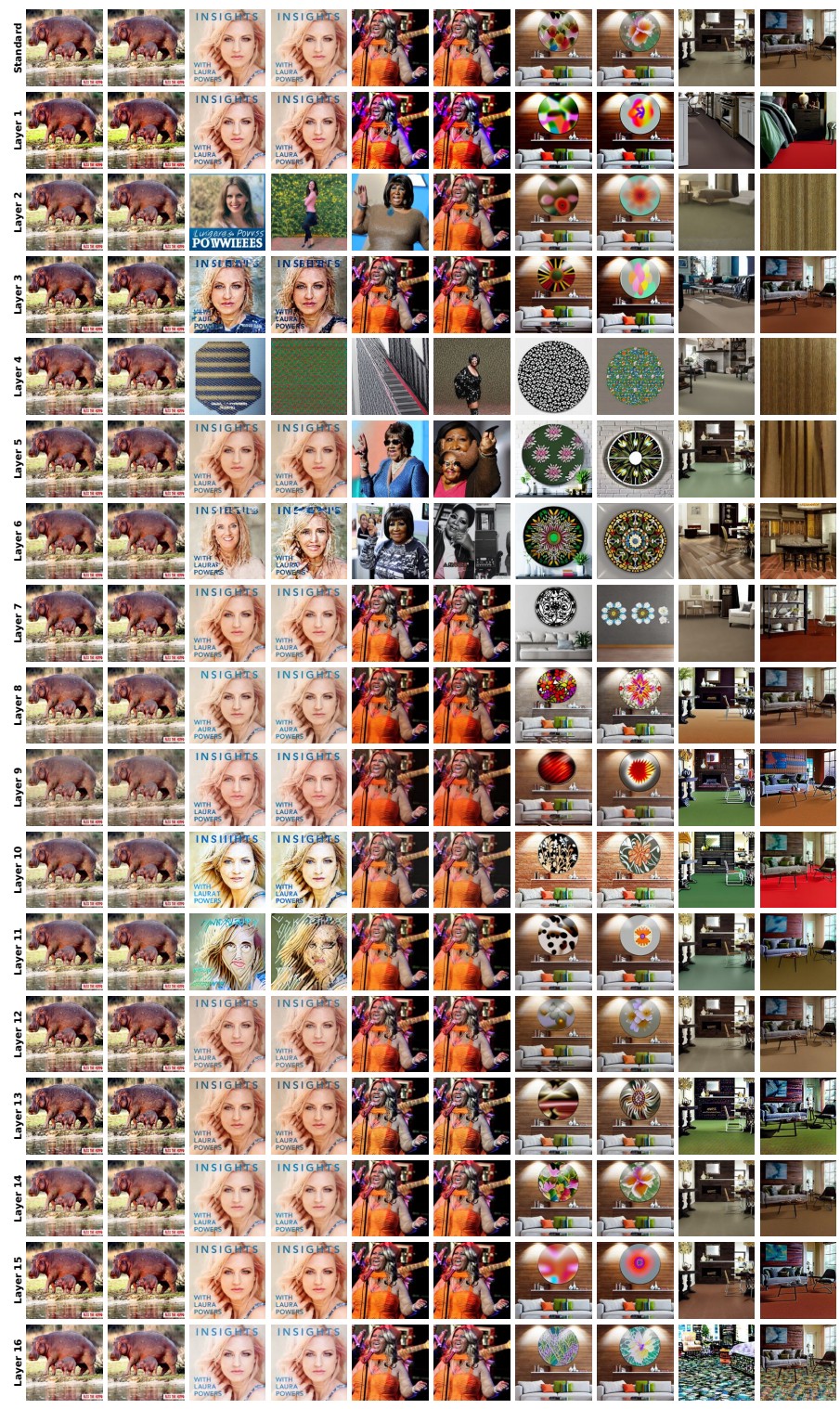

Figure 19: **Images generated with memorized prompts with deactivated individual key layers.** Whereas the *standard* row shows generations with keeping all neurons active, the following rows depict results for deactivating all neurons in a specific key layer.

## C.9 Ablation of Individual Convolutional Layers

We also experimented with deactivating neurons in other U-Net layers, including both convolutional and fully connected layers, but we did not find any indicators of memorization in these units. Even when deactivating numerous neurons in the convolutional and fully connected layers of the U-Net, the memorized training images were still faithfully reproduced. However, the quality of the images degraded, particularly when deactivating neurons in early layers, which are responsible for defining the image structure. We showcase in Fig. 20 various examples of memorized images generated with $50\%$ deactivated neurons in the convolutional layers to illustrate that these neurons have no noticeable impact on the memorization behavior. However, deactivating too many neurons in early layers can negatively affect the overall generation process. So, to conclude, deactivating other neurons in the U-Net didn't seem to impact memorization, which is why our choice to focus on the value layers seems reasonable.

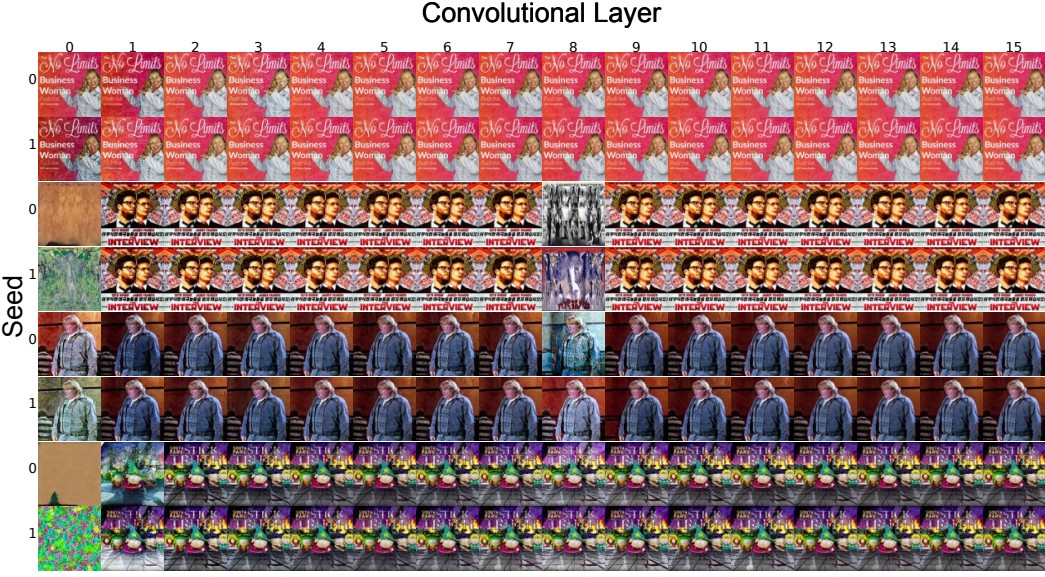

Figure 20: For each convolutional layer in the U-Net's down blocks, we randomly deactivated 50% of the neurons. The results demonstrate that blocking neurons in the convolutional layers does not mitigate memorization. Instead, deactivating the neurons reduces the quality of the generated images and, in some cases, causes the entire generation process to collapse, especially when neurons in the early layers are deactivated.

# D  Algorithmic Description of NeMo

## D.1  Computing Noise Differences

Alg. 1 defines our algorithm to compute the differences between the initial noise samples and the noise predicted during the first denoising step. The resulting noise differences are used to compute our SSIM-based memorization score during the initial neuron selection and the refinement step. We compute the noise differences always for $n = 10$ different seeds to avoid undesired biases due to the random sampling process. We further remove noise differences from the set, which have low similarity with other noise differences. By this step, we remove noise differences for seeds that do not lead to memorization and might mislead the algorithm.

---

**Algorithm 1** Compute Noise Differences

---

**Input:**
    Prompt embedding $y$                                            $\triangleright$ Text prompt (embedding)
    Neuron set $S_{\text{neurons}}$                              $\triangleright$ Set of neurons to deactivate
    Noise predictor $\epsilon_\theta$                                    $\triangleright$ Diffusion model
    Memorization threshold (SSIM) $\tau_{\text{mem}}$           $\triangleright$ Target memorization score

**Output:**  Noise differences $\Delta$

    Set $\Delta$ as empty list                           $\triangleright$ Initialize list of noise differences
    $\tilde{\epsilon}_\theta \leftarrow$ deactivate_neurons$(\epsilon_\theta, S_{\text{Neurons}})$     $\triangleright$ Set activations of neurons in $S_{\text{neurons}}$ to zero

    *// Compute noise differences for each random seed*
    **for** $i = 1, \ldots, 10$ **do**                              $\triangleright$ Iterate over 10 seeds
        set_seed(i)                               $\triangleright$ Set random seed to $i$
        sample $x_T \sim \mathcal{N}(\mathbf{0}, \mathbf{I})$            $\triangleright$ Randomly initialize noise image
        $x_{T-1} \leftarrow \tilde{\epsilon}_\theta(x_T, T, y)$            $\triangleright$ Compute noise prediction
        $\delta \leftarrow x_{T-1} - x_T$                 $\triangleright$ Compute noise difference
        $\delta \leftarrow \frac{\delta - \min(\delta)}{\max(\delta) - \min(\delta)}$        $\triangleright$ Normalize differences by min-max scaling
        append $\delta$ to $\Delta$              $\triangleright$ Add current noise difference to list
    **end for**

    *// Remove noise differences not leading to memorization*
    **for** $\delta \in \Delta$ **do**                            $\triangleright$ Iterate over noise differences
        $\bar{\Delta} \leftarrow \Delta \setminus \delta$                $\triangleright$ Get set of noise differences without $\delta$
        $d \leftarrow$ compute_memorization$(\delta, \bar{\Delta})$     $\triangleright$ Compute pairwise memorization scores (SSIM)
        **if** $\max(d) < \tau_{\text{mem}}$ **then**        $\triangleright$ Highest memorization score is below threshold
            $\Delta \leftarrow \Delta \setminus \delta$            $\triangleright$ Remove noise difference from set
        **end if**
    **end for**
    **return** $\Delta$                              $\triangleright$ Return list of noise differences

---

## D.2   Detecting Neurons with Out-of-Distribution Activations

Alg. 2 describes our method to detect neurons with out-of-distribution (OOD) activations. Our method detects OOD neurons based on their activation distance for a memorized prompt to a neuron's mean activation computed on a hold-out dataset of non-memorized prompts. In addition, we also add the $k$ neurons with the highest absolute activations within each layer to the set.

---

**Algorithm 2** Get OOD Neurons

---

**Input:**

    Prompt embedding $y$                                                   $\triangleright$ Text prompt (embedding)

    Activation threshold $\theta_{\text{act}}$                                  $\triangleright$ Threshold for the OOD detection

    Top $k$                                                      $\triangleright$ Value of top-k detection

    Activation mean $\mu$ and standard deviation $\sigma$      $\triangleright$ Activation statistics of hold-out dataset

**Output:**   Set of neurons with OOD activations $S_{\text{initial}}$

    $S_{\text{activations}} \leftarrow \text{collect\_activations}(y)$                   $\triangleright$ Collect activations on prompt

    $S_{\text{initial}} \leftarrow \{\}$                                     $\triangleright$ Initialize empty neuron set

    *// Check each neuron in each layer for OOD activation*

    **for** $l \in \{1, \ldots, L\}$ **do**                             $\triangleright$ Iterate over all layers

        **for** $i \in \{1, \ldots, N\}$ **do**               $\triangleright$ Iterate over all $N$ neurons in layer $l$

            $z_i^l(y) = \frac{a_i^l(y) - \mu_i^l}{\sigma_i^l}$          $\triangleright$ Compute z-score for current neuron

        **if** $z_i^l(y) > \theta_{\text{act}}$ **then**             $\triangleright$ Activation above OOD threshold

            $S_{\text{initial}} \leftarrow S_{\text{initial}} \cup \{\text{neuron}_i^l\}$      $\triangleright$ Add OOD neuron to set

        **end if**

        **end for**

    *// Add k neurons of layer l with the highest absolute activations to the candidate set*

        $S_{\text{topk}} \leftarrow \text{top\_k\_activations}(S_{\text{activations}}, l, k)$    $\triangleright$ Get neurons with highest absolute activations

        $S_{\text{initial}} \leftarrow S_{\text{initial}} \cup S_{\text{topk}}$             $\triangleright$ Add top-k neurons to set

    **end for**

    **return** $S_{\text{initial}}$                                $\triangleright$ Return set with OOD neurons

---

## D.3 Selecting Initial Candidates of Memorization Neurons

Alg. 3 defines our algorithm to compute the initial set of memorization neurons. The resulting initial set of selected memorization neurons is then refined in a second step, shown in Alg. 4.

---

**Algorithm 3** Initial Neuron Selection

---

**Input:**

    Prompt embedding $y$                                             ▷ Text prompt embedding

    Memorization threshold (SSIM) $\tau_{\text{mem}}$                     ▷ Target memorization score

    Minimum activation threshold $\theta_{\text{min}}$                  ▷ Threshold for stopping neuron search

**Output:**    Set of neuron candidates $S_{\text{initial}}$, refinement memorization threshold $\tau_{\text{mem\_ref}}$

    Candidate set of memorization neurons $S_{\text{initial}}$              ▷ Initial memorizing neuron set

    Memorization threshold (SSIM) $\tau_{\text{mem\_ref}}$            ▷ Memorization threshold for refinement

    $\text{mem} \leftarrow 1.0$                           ▷ Initialize memorization score as maximum

    $\theta_{\text{act}} \leftarrow 5$                           ▷ Initialize threshold of OOD activation detection

    $k \leftarrow 0$                              ▷ Initialize $k$ for top-$k$ activation detection

    $\tau_{\text{mem\_ref}} \leftarrow \tau_{\text{mem}}$               ▷ Set refinement memorization threshold to current threshold

    $\Delta_{\text{unblocked}} \leftarrow \text{get\_noise\_diff}(y, \emptyset)$          ▷ Noise differences with all neurons active

    *// Increase set of candidate neurons until target memorization score is reached*

    **while** $\text{mem} > \tau_{\text{mem}}$ **do**               ▷ While memorization score above threshold

        $S_{\text{initial}} \leftarrow \text{get\_ood\_neurons}(y, \theta_{\text{act}}, k)$       ▷ Detect neurons with OOD activations

        $\Delta_{\text{blocked}} \leftarrow \text{get\_noise\_diff}(y, S_{\text{initial}})$          ▷ Compute noise differences

        $\text{mem} \leftarrow \text{compute\_memorization}(\Delta_{\text{unblocked}}, \Delta_{\text{blocked}})$ ▷ Compute memorization score (SSIM)

        **if** $\theta_{\text{act}} < \theta_{\text{min}}$ **then**              ▷ Minimum activation threshold not reached

            $\tau_{\text{mem\_ref}} \leftarrow \text{mem}$        ▷ Set refinement threshold to current memorization score

            **break**                    ▷ Stop if activation threshold is too low

        **end if**

        *// Adjust OOD detection parameters to increase set of candidate neurons*

        $\theta_{\text{act}} \leftarrow \theta_{\text{act}} - 0.25$            ▷ Decrease threshold for OOD detection

        $k \leftarrow k + 1$                   ▷ Increase $k$ for top-$k$ activation detection

    **end while**

    **return** $S_{\text{initial}}, \tau_{\text{mem\_ref}}$       ▷ Return neuron candidates and refinement memorization threshold

---

### D.4 Neuron Selection Refinement

Alg. 4 defines our algorithm to refine the set of candidate neurons identified from NEMO's initial selection step.

---

**Algorithm 4** Neuron Selection Refinement

---

**Input:**
    Initial memorization neuron candidate set $S_{\text{initial}}$         ▷ Given neuron candidate set
    Memorization threshold (SSIM) $\tau_{\text{mem\_ref}}$       ▷ Refinement memorization score threhsold

**Output:**   memorization neurons $S_{\text{refined}}$          ▷ Refined set of memorization neurons

    $S_{\text{refined}} \leftarrow S_{\text{initial}}$
    $\Delta_{\text{unblocked}} \leftarrow \text{get\_noise\_diff}(y, \emptyset)$       ▷ Noise differences with all neurons active

    *// Check all candidate neurons of individual layers at once for memorization*
    **for** $l \in \{1, \ldots, L\}$ **do**         ▷ Iterate over all layers to remove low impact layers
        $S_{\text{layer}} \leftarrow \text{get\_neurons\_in\_layer}(S_{\text{refined}}, l)$      ▷ Get the neurons in the current layer $l$
        $S_{\text{neurons}} \leftarrow S_{\text{refined}} \setminus S_{\text{layer}}$      ▷ Compute set of neurons from remaining layers
        $\Delta_{\text{blocked}} \leftarrow \text{get\_noise\_diff}(y, S_{\text{neurons}})$       ▷ Compute noise differences
        $\text{mem} \leftarrow \text{compute\_memorization}(\Delta_{\text{unblocked}}, \Delta_{\text{blocked}})$ ▷ Compute memorization score (SSIM)

        **if** $\text{mem} < \tau_{\text{mem\_ref}}$ **then**       ▷ Minimum memorization threshold not reached
           $S_{\text{refined}} \leftarrow S_{\text{refined}} \setminus S_{\text{layer}}$      ▷ Remove neurons of layer $l$ from neuron set
        **end if**
    **end for**

    *// Check all remaining candidate neurons individually*
    **for** $l \in \{1, \ldots, L\}$ **do**         ▷ Iterate over each remaining layer
        $S_{\text{layer}} \leftarrow \text{get\_neurons\_in\_layer}(S_{\text{refined}}, l)$      ▷ Get the neurons in the current layer $l$

        **for** $n \in S_{\text{layer}}$ **do**
           $S_{\text{neurons}} \leftarrow S_{\text{refined}} \setminus \{n\}$      ▷ Compute set of neurons without neuron $n$
           $\Delta_{\text{blocked}} \leftarrow \text{get\_noise\_diff}(y, S_{\text{neurons}})$       ▷ Compute noise differences
           $\text{mem} \leftarrow \text{compute\_memorization}(\Delta_{\text{unblocked}}, \Delta_{\text{blocked}})$    ▷ Compute mem. score (SSIM)

           **if** $\text{mem} < \tau_{\text{mem\_ref}}$ **then**       ▷ Minimum memorization threshold not reached
               $S_{\text{refined}} \leftarrow S_{\text{refined}} \setminus \{n\}$      ▷ Remove current neuron from set
           **end if**
        **end for**
    **end for**

    **return** $S_{\text{refined}}$          ▷ Return refined set of memorization neurons

---

