# OpenReview forum: "Finding NeMo: Localizing Neurons Responsible For Memorization in Diffusion Models"
_NeurIPS.cc/2024/Conference — NeurIPS 2024 poster_

### Official Review · Reviewer_RyXV · 2024-07-10

**Soundness:** 3
**Presentation:** 3
**Contribution:** 2
**Rating:** 6
**Confidence:** 3

**Summary:**

This paper introduces NeMo, a straightforward and easy-to-understand method. NeMo identifies neurons responsible for memorization and deactivates them, offering potential applications in the protection of sensitive or copyrighted data.

**Strengths:**

This topic is highly interesting as it enhances interpretability and offers a lightweight solution for addressing issues related to sensitive or copyrighted training images.

**Weaknesses:**

1, There is insufficient comparison with other methods, such as the work mentioned in [1]. The authors state, "While mitigations that rely on preventing the generation of memorized samples are effective when the DM is developed and deployed in a secure environment, they hold the inherent risk of adversaries circumventing them. Additionally, they are not effective when the DMs are publicly released, such that users can freely interact with them." However, this argument is not persuasive enough. Providing more compelling reasons for the unique position of this method in mitigating memorization would enhance my evaluation score.

2, The side effects of this method are not thoroughly investigated.


[1] Detecting, Explaining, and Mitigating Memorization in Diffusion Models

**Questions:**

1, I have some reservations about the connection between copyright/sensitive images and memorization. It seems that copyright issues might only be partially triggered by memorization. Could you provide further clarification on this point? This pertains to the application of your method.

2, I find the activation-based localization method reasonable. However, I'm curious if pruning methods such as [1] or other neuron localization methods such as [2] could also be effective here. I would appreciate some analysis on this, though I do not require any additional experiments.

3, In section 4.3, could you explain why deactivating neurons further reduces FID and KID in some cases?

4, I'm also interested in the potential side effects of this method. You mention that deactivation has a minimal impact on non-memorized generation, which suggests that these neurons do not interact significantly with other neurons and function independently. I believe a more detailed analysis from this perspective would be beneficial.


[1] Structural Pruning for Diffusion Models
[2] Cones: Concept Neurons in Diffusion Models for Customized Generation.

**Limitations:**

The limitations have been discussed.

---

> ### Author Rebuttal · Authors · 2024-08-07
>
> **Providing more compelling reasons for the unique position of this method in mitigating memorization.**
> > The novelty and positioning of our research is multi-faceted: NeMo is the first method to localize memorization in diffusion models down to individual neurons. Previous research has generally focused on detecting memorized prompts in general. Still, no method has yet offered a way to identify specific parts of the model responsible for this memorization behavior.
>
> > By identifying memorizing neurons, one can prune these neurons, permanently removing them from the model. This enables model providers to eliminate undesired memorization before releasing the model to the public, preventing users from simply deactivating the mitigation technique. Existing mitigation methods rely on embedding or attention adjustments at inference time, which users can easily deactivate, thus failing to offer a permanent solution.
>
> > Unlike traditional model pruning literature, we aim not to minimize the total number of model parameters without performance loss but to identify those parameters responsible for memorizing particular training samples.
>
> **Comparison to research on pruning and concept neurons for diffusion models.**
> > Structural pruning for diffusion models [1] aims to reduce the number of model parameters by removing sub-structures that can be eliminated without hurting the model's performance. The goal of structural pruning is orthogonal to ours. While structural pruning focuses on reducing the parameter count, it does not address sample memorization. Its optimization criterion is based on the standard reconstruction loss used during the training of diffusion models. Although one might adapt this approach to mitigate memorization, such adjustments would require additional data and resource-intensive training. In contrast, our method identifies memorization neurons using a gradient-free approach and removes only these neurons to mitigate memorization. It requires no additional training or data, acting solely on input prompts.
>
> > Concept neurons [2] is a gradient-based method designed to identify neurons associated with specific image concepts for customized image generation. Their method requires several images of the target concept with sufficient variations. Concept neurons and our approach differ significantly in their objectives and methodologies. Concept neurons focus on identifying neurons associated with a concept across multiple images, making this method unsuitable for localizing memorization. Furthermore, their approach necessitates fine-tuning the diffusion model on concept images before identifying neurons based on gradient information. In contrast, our method identifies memorization neurons responsible for individual prompts.
>
> > We thank the reviewer for pointing out the potential connection between our work and these two papers, which offer interesting avenues for future research into memorization. We will include a discussion on the relation of our approach to the pruning literature in our paper.
>
> **Clarifying the connection between copyright/sensitive images and memorization?**
>
> > We are happy to elaborate further on this point. Memorization, particularly verbatim memorization, which reconstructs training samples perfectly, can pose significant copyright issues. For instance, the generated images in Figure 3, which (without mitigation) reproduce training samples exactly, may include content that is not license-free. Users might not realize that their generated images closely resemble copyrighted material, leading to legal issues due to copyright violations.
>
> > Moreover, some memorized images may correspond to training data that users have unintentionally made publicly accessible. While the content described by the corresponding prompt may not be inherently sensitive, the exact replication of the training sample is. In such cases, removing the identified memorizing neurons provides a practical solution to these issues without requiring time-consuming and computationally expensive re-training or fine-tuning.
>
>
> **Could you explain why deactivating neurons further reduces FID and KID in some cases?**
>
> > We hypothesize there are two reasons for the reduction in FID/KID. First, deactivating the memorization neurons increases the variation in generated images with (partly) memorized contents. This effect might also (slightly) improve the overall image quality of non-memorized prompts. Second, there is some noise when computing the quality metrics, as indicated by the standard deviations, which might also explain the improvements. Interestingly, the Structural pruning for diffusion models [1] paper also reports improvements in FID score when pruning a CelebA-HQ model (Table 1 in their paper). Further analyzing the connection between pruning and image quality promises an exciting avenue for future research. The important take-away message for our work is that removing memorization neurons does not hurt the model's general performance.
>
> **More detailed analysis on side effects and neuron functionality.**
> > We ran additional experiments to analyze the impact and effects of deactivating neurons in the value layers. We deactivated a varying number of neurons, either randomly selected or the top memorizing neurons, and measured the similarity between the outputs of the cross-attention blocks with deactivated neurons and all neurons activated. Our results, stated in the rebuttal PDF, show that even deactivating a large number of neurons only impacts the attention outputs marginally. We, therefore, conclude that other neurons can replace the functionality of specific neurons on non-memorized prompts and that neurons in the value layers act rather independently. Yet, for neurons memorizing specific prompts, this memorizing functionality is not replaced by other neurons, explaining the mitigation effect of deactivating these neurons.

---

> > ### Comment · Reviewer_RyXV · 2024-08-11
> >
> > I appreciate the authors' detailed rebuttal and additional insights. Most of my questions have been addressed, and I will be maintaining my original rating.

---

> > > ### Author Response · Authors · 2024-08-12
> > >
> > > Thank you for your detailed rebuttal. We're glad that you are leaning towards accepting the paper. If you have any further questions or comments, we would be happy to provide additional information.

---

### Official Review · Reviewer_K7EP · 2024-07-11

**Soundness:** 3
**Presentation:** 3
**Contribution:** 3
**Rating:** 6
**Confidence:** 5

**Summary:**

This paper introduces NEMO, a method that identifies and deactivates specific neurons responsible for memorizing training data in DMs. This approach prevents the reproduction of sensitive images, enhances output diversity, and mitigates data leakage, enabling more responsible use of DMs.

**Strengths:**

1. It explains the connection between neurons and memorization, which provides more understanding in this problem.

2. This paper provides two selection stages to select the different neurons.

3. The paper writing is clear.

**Weaknesses:**

1. The author should specify their novelty compared with [1]. The novelty seems limited considering the conclusion in [1].

2. No practical mitigation method is provided. Deactivating is not possible without knowing the neurons.

3. The author claim that most memorization happens in the value mappings of the cross-attention layers of DMs. However, actually, the author only prove that some neurons in attention layers are necessary in memorization. They didn't show that those neurons are everything about the memorization. It is possible that deactivating other neurons that are not in attention layers can also remove memorization. Localizing should consider how to exclude other neurons.

**Questions:**

The author should address weakness 1 and 3.

**Limitations:**

The limitations that the author mentioned in Appd A is about how to further improve the method. It is not clear solved but propose the key factors is possible to solve in the future work.

---

> ### Author Rebuttal · Authors · 2024-08-06
>
> **The author should specify their novelty compared with [1]. The novelty seems limited considering the conclusion in [1].**
>
> > We believe there is a reference missing here in the review. Could you please specify which paper we should compare our method to? Reference [1] in our paper does not seem to correspond to the mentioned work. Additionally, we emphasize that we already compare our method to the current state-of-the-art mitigation method by Wen et al. (ICLR 2024) in Section 4.1 and Table 1, as well as to the most effective mitigation strategy proposed by Somepalli et al. (NeurIPS 2023) in Appendix C.9 and Table 3. We also added additional results for the mitigation proposed by Ren et al. (Preprint 2024) in Table 2 in the attached rebuttal PDF. Please let us know if this already addresses your concern or to which paper citation [1] refers.
>
> **No practical mitigation method is provided. Deactivating is not possible without knowing the neurons.**
>
> > There seems to be a misconception here. Our method is explicitly designed to identify and localize the neurons responsible for memorizing training data. For each memorized prompt, we identify a set of neurons that can then be deactivated to mitigate memorization. Indeed, deactivating neurons is not feasible without first identifying them, which is why our method focuses on this identification process. Therefore, we respectfully disagree with the notion that our method is impractical. We are happy to address any further questions or clarifications that may arise.
>
> **The authors prove that some neurons in the attention block's value layers are necessary for memorization. What about other layers?**
>
> > We kindly refer to our analysis in Appx. C.8, where we analyzed the memorization behavior of other neurons in the network. In our paper, we focused on neurons that directly process the text embeddings and are responsible for guiding the image generation process. Focusing on these layers is reasonable since memorization directly depends on the concrete input prompt, as we have shown in our analysis.
>
> > In Stable Diffusion, the key and value layers process the text embeddings of the prompt. Therefore, we conducted experiments showing that the value layers are responsible for memorization in diffusion models. The neurons in the key mapping layers directly process the text embeddings to compute the attention scores. However, strong interdependencies exist between the activations of different neurons through the nature of the softmax function. The impact each neuron’s activation has on the computed attention score also depends on the activations of all other neurons from the same layer. Removing a single neuron, i.e., setting its activation to zero, does not necessarily imply substantial changes in the attention scores and the corresponding weighting of features from the value mapping layer. The value mapping layers, however, also directly process the text embeddings, but there is no direct interdependence between the activations of different neurons.
>
> > Consequently, setting the activations of individual neurons in the value layers to zero directly blocks the information flow from the text embeddings. Therefore, it is reasonable to focus on the neurons in the value layers to mitigate memorization. We also experimented with deactivating neurons in other U-Net layers, including both convolutional and fully connected layers, but we did not find any indicators of memorization in these units. Even when deactivating numerous neurons in the convolutional and fully connected layers of the U-Net, the memorized training images were still faithfully reproduced. However, the quality of the images degraded, particularly when deactivating neurons in early layers, which are responsible for defining the image structure. We showcase in the rebuttal PDF various examples of memorized images generated with 50\% deactivated neurons in the convolutional layers to illustrate that these neurons have no noticeable impact on the memorization behavior. However, deactivating too many neurons in early layers can negatively affect the overall generation process. So to conclude, deactivating other neurons in the U-Net didn't seem to have an impact on memorization which is why our choice to focus on the value layers seems to be a reasonable choice.

---

> > ### Comment · Reviewer_K7EP · 2024-08-09
> > **Missed reference and follow-up response**
> >
> > Sorry about the missed reference. Please compare the novelty with Unveiling and Mitigating Memorization in Text-to-image Diffusion Models through Cross Attention. Since they also find the attention score may show connection with memorization. The contribution of this paper seems limited.
> >
> > I disagree with your reponse to weakness 3. From the end of input to the end of output, the models are structued by layer by layer and module by module. Value may memorize text embedding, but the following layer must also memorize the output of attention layer. Otherwise the final output cannot be memorized. It is like a chain. Memorization will not work without the any one of the layer. The memorization is never something that is totally dependent on one neuron or a few neurons. In other words, these neurons are just some clues or signals of memorization. It is not correct to say they control memorization. It is correct to say value or kay layers has a very special behavior that is related to memorization. But they didn't decides the whole thing. In other layers like the convolutional layers in u-net, they must also memorize their input. That memorization is not probabily about one neuron. It is maybe need many neurons for a memorization. The author cannot say something they didn't find does not exist. If the author acknowledge this and update in their future version, I will increase my score. Otherwise I would possibily decrease my score.

---

> > > ### Comment · Reviewer_K7EP · 2024-08-09
> > >
> > > I would suggest to list your response. Otherwise it would increase the cost to write the response for reviewers, which may decrease the possibility to get a response.

---

> ### Author Response · Authors · 2024-08-06
> **Missing Reference**
>
> Dear ACs and PCs,
> dear Reviewer
>
> due to a lack of reference [1] in the review, we were not able to provide the comparison requested by the reviewer during the rebuttal phase. We would have been happy to provide the requested information, but this was not possible during the initial rebuttal phase without further details from the reviewer.
>
> We, therefore, kindly ask the Reviewer to provide additional information.
>
> Best regards,
> the authors

---

> ### Comment · Area_Chair_qTny · 2024-08-08
> **Can you specify the missing reference?**
>
> Dear Reviewer K7EP,
>
> Can you specify the missing reference [1] in your review? It seems to have been dropped in your original review. I'm not sure if there will be enough time for the authors to provide comparison but at least some clarifications regarding the novelty could be discussed during this week. Thanks!
>
> AC

---

> ### Author Response · Authors · 2024-08-10
>
> Dear Reviewer K7EP,
>
> thank you for your quick response and the clarification of the missing reference.
>
> **1.) The contribution of this paper seems limited given that Ren et al. show that memorization and cross attention may have a connection**: We first emphasize that the referred paper (Ren et al.) is an unpublished preprint made available on Arxiv about 2 months (March 17th) before the NeurIPS deadline, such that it would fulfill the definition of concurrent work. Additionally, a thorough comparison between the two papers yields that they are conceptually different and that our work has substantially novel contributions:
> - a) Ren et al. investigate the attention scores, i.e., the distribution of the dot product between key and query vectors, finding that attention concentrates more on the beginning token for non-memorized prompts and is more concentrated on certain trigger tokens for memorized prompts. Whereas this is an interesting finding, it is orthogonal to our work. While Ren et al. investigate the model's input space in terms of CLIP embeddings for memorization, we focus on the U-Net itself and identify the neurons in the value mapping responsible for triggering memorization.
> - b) The findings of Ren et al. are more coarse grained (layer focus rather than neuron focus), which enables them to make a related finding to ours. Still, they do not consider the most interesting part of our research, namely that most of the memorization is triggered by 1 or a handful neurons in the value layers.
> - c) The inference-time mitigation strategy by Ren et al. is conceptually different from ours. Based on their findings 1 and 2 (both on the input level), where they look at token entropy, their method consists of masking out the summary tokens and increasing the logits of the beginning token. This strategy is very different from ours, which prunes individual neurons from the value mapping in cross-attention layers. Also, their training time mitigation simply removes samples with high attention entropy, which is somewhat related to our detection of out-of-distribution activation in the value layers. Yet, our mitigation strategy acts at inference time, a more realistic and practical scenario given the amount of computing required for training text-to-image models.
>
> We, therefore, kindly disagree that our paper's contribution is limited. We also emphasize that the other three Reviewers all agree on the novelty and importance of our research and findings.
>
> **2.) Memorization in Other Layers**: It seems there might have been some misunderstanding between the question raised and our response. We fully acknowledge that layers following the value mappings in the U-Net also play a significant role in memorizing training samples. The convolutional and fully connected layers, being integral to the process of constructing images by denoising the initial input, naturally contribute to the memorization of the appearance of training samples.
>
> However, the focus of our paper is on identifying the specific signals that trigger the generation and reproduction of memorized data in the first place. Our findings indicate that certain neurons in the value layers of the cross-attention blocks act as such triggers. We do not claim that these neurons are solely responsible for memorization or that other model components are irrelevant to the process. Our research specifically aims to detect these triggering parts within the model, and our analysis identified them in the value layers of cross-attention blocks.
>
> While we also analyzed the convolutional layers, we did not find individual channels exhibiting similar trigger behavior, where removing such channels would mitigate memorization. This does not imply that these layers are unimportant for reconstructing memorized samples. We are happy to clarify this point in our paper.

---

> > ### Comment · Reviewer_K7EP · 2024-08-11
> >
> > Thanks for your response. For the second question, I would say before K and V, there are also some components that would play some roles in memorization. For example, transformations are conducted from text embedding space to the input space of K and V. That is to say, the memorization is like a chain, while the attetion neurons are one part in the middle. Since it is in the middle, it is not correct to say they trigger memorization. They are only the one that is found to have special behaviors which are connected with memorization. They are actually triggered by their input.

---

> > > ### Author Response · Authors · 2024-08-12
> > >
> > > Thank you for pointing that out. We agree that the input prompt serves as the actual initiation of the memorization process, and without a specifically memorized prompt, the model is unlikely to trigger memorization. In this regard, transforming the prompt into an embedding vector is critical and marks the start of this process. If the text encoder would transform the prompt into a different embedding—for instance, by deactivating parts of the model or through fine-tuning—memorization might not occur in the output of the diffusion model. However, as noted in our previous response, we interpret the text embedding as a fixed conditioning factor since the text encoder (CLIP for Stable Diffusion) was trained separately using a contrastive method (on OpenAI’s internal data) and likely never saw the exact prompts memorized by the diffusion model (LAION 2B) during its training. In this scenario, the processing of the text embedding begins with the key and value layers in the U-Net, marking the start of the memorization chain within the U-Net, which is triggered by the input embedding.
> > >
> > > We will incorporate the perspective and discussion of the memorization chain, starting from the input prompt, into our final paper. We hope this addresses all your questions and concerns regarding the memorization aspect.

---

> > > > ### Comment · Reviewer_K7EP · 2024-08-12
> > > >
> > > > I have raised my score.

---

> > > > > ### Author Response · Authors · 2024-08-12
> > > > >
> > > > > Thank you for your detailed rebuttal and for the engaging and fruitful discussion. Your insights will definitely enhance our paper. We're also pleased to see that you have raised your score in favor of accepting the paper.

---

### Official Review · Reviewer_zaL8 · 2024-07-11

**Soundness:** 2
**Presentation:** 2
**Contribution:** 2
**Rating:** 6
**Confidence:** 4

**Summary:**

The paper localize memorized samples to neurons in the cross attention layers in diffusion models. By deactivating neurons responsible for memorization, the proposed method enables models to generate diverse images different from training images.

**Strengths:**

1. The authors propose the first method to localize memorization within the model, which is a novel idea that has not been studied in the context of text-to-image generation models before.
2. The experiments and findings presented in the paper are interesting.
3. The paper is well-written and easy to follow.

**Weaknesses:**

1. I understand that detecting memorized propmts is out of scope of this paper. Previous methods in mitigating memorization construct an entire pipeline in which memorized prompts are first detected with a heuristic/metric, and a mitigation strategy is triggered during inference/training if the metric surpasses a certain threshold. My expectation from "localization of memorization" was that there is a specific subset of neurons that, when blocked, avoid replication of training data. Such a localization would not require the authors to develop a pipeline to detect memorization as those neurons can be pruned.

(a) From the paper (Table 1 and Figure 4) it seems that the proposed method operates on a per-prompt basis and all experiments have been conducted by separating the memorized prompts into template and verbatim.

(b) I believe that although these findings are interesting, one weakness is that the method is impractical as in practice, it would still require one to know the type of memorization before deciding which neurons to block. For some highly memorized prompts, the highly-iterative neuron discovery process (Alg D.1 to D.4) needs to be repeated. This not only requires interference with the inference pipeline but is also computationally inefficient.

**Questions:**

1. How are tokens handled while calculating neuron activation in Equation 3? Do you consider an average or max over tokens?
2. How faithful is SSIM for detecting memorized prompts? Can it be used to detect if a prompt is memorized or not?

**Limitations:**

The authors have clearly addressed the limitations of the proposed method in the paper. I believe some more limitations are addressed in the Weaknesses section.

---

> ### Author Rebuttal · Authors · 2024-08-06
>
> **My expectation from "localization of memorization" was that there is a specific subset of neurons that, when blocked, avoid replication of training data. Such a localization would not require the authors to develop a pipeline to detect memorization, as those neurons can be pruned.**
>
> > It would indeed be very helpful to identify a fixed set of neurons responsible for all types of memorization in a model beforehand, without the need to first identify memorized prompts individually. However, as our analyses demonstrate, such a fixed set of memorization neurons does not exist in diffusion models. Instead, we identified various neurons responsible for the memorization of different prompts. While some sets of neurons memorize multiple prompts, no fixed set of neurons is responsible for memorizing all prompts. This is not a weakness of our method, but rather a reflection of the behavior of already trained models.
>
> **It seems that the proposed method operates on a per-prompt basis, and all experiments have been conducted by separating the memorized prompts into template and verbatim.**
>
> > There appears to be a misunderstanding here. We only split the results into template and verbatim memorization to provide better insights into the effects of different types of memorization. Our algorithm and all hyperparameters are fully agnostic to the type of memorization. We applied our method to all memorized prompts from our test set in the exact same way, without setting any memorization- or prompt-specific hyperparameters. We used the same values and thresholds for all memorized prompts. The distinction between memorization types is only made when presenting our results to support our analyses and offer additional insights. We will gladly clarify this point in our final version.
>
> **The method is impractical, as it requires one to know the type of memorization before deciding which neurons to block.**
>
> > As discussed in the previous answer, there seems to be a misunderstanding here. Our method is fully agnostic to the type of memorization, so in practice, one does not need to know the type of memorization to apply it. Furthermore, as detailed in our ablation study and sensitivity analysis in Appendix C.7, the user does not even need to know if a prompt is memorized at all. In the sensitivity analysis, we applied our method to non-memorized prompts without any adjustments and found that for the majority of these prompts, our method detected no memorization neurons. This is expected since these prompts show no memorization behavior. For the remaining non-memorized prompts, a median of 62 neurons was identified. Deactivating these neurons should not impact the image quality, as our analysis in Section 4.4 and Fig. 5a) demonstrates.
>
> **For some highly memorized prompts, the neuron discovery process needs to be repeated. This not only requires interference with the inference pipeline but is also computationally inefficient.**
>
> > We emphasize that our detection method only needs to be run once for each input prompt. While the runtime varies based on the strength of the memorization, for verbatim memorized prompts, which represent the strongest type of memorization, the runtime was only a few seconds (as discussed in our limitations section in Appendix A). Once we have identified the memorizing neurons, we can permanently deactivate them. This allows future inference steps to generate images without the need to repeat the localization process for the memorized prompts or interfere with the inference pipeline at any point. In contrast, related inference-time mitigations need to be applied for each new generation, increasing the total runtime and memory consumption of the generation process. Therefore, we kindly disagree that our method is computationally inefficient.
>
>
> **How are tokens handled while calculating neuron activation in Equation 3? Do you consider an average or max over tokens?**
>
> > We did not interact with or adjust any tokens of the input prompts. The model's text encoder first processes the prompt—the CLIP text encoder in the case of Stable Diffusion—and transforms it into embedding vectors of a fixed size ($77\times 768$), independent of the prompt length (by padding the input tokens). In Equation 3, we only consider the computed embedding vectors as input to the i-th neuron in the l-th layer, denoted by $a^l_i(y)$. To quantify neuron activation, we take the mean of the absolute activations across all 77 activations in the embedding vectors. We will clarify this computation in our final version.
>
> **How faithful is SSIM for detecting memorized prompts? Can it be used to detect if a prompt is memorized or not?**
>
> > Indeed, the SSIM score itself can be used to detect memorization. We visualized the different SSIM score distribution for non-memorized and memorized prompts in Fig. 2a) in the main paper. To underline this observation with quantitative metrics, we ran additional experiments to explore the detection capabilities of our SSIM-based method. We measured the efficiency of the SSIM score for memorization detection on our dataset of memorized and non-memorized prompts. Without extensive hyperparameter tuning, this detection method achieves an AUROC of 98\%, an accuracy of 94.2\% (using a naive threshold of 0.5 on the SSIM similarity), and a TPR@1\%FPR of 87.6\%. Since the amount of memorization of template memorized prompts varies significantly, we repeated the computation for detecting the verbatim memorized prompts. Here, the SSIM approach even achieves an AUROC of $99.64\%$, an accuracy of $97.39\%$ (with a threshold of 0.5), and a TPR@1\%FPR of 98.21\%. These results indicate that the SSIM score can also be used to detect memorization reliably in the first place. However, detection is not the focus of the paper but the localization on the neuron level.

---

> > ### Comment · Reviewer_zaL8 · 2024-08-09
> > **Thank you for the rebuttal**
> >
> > Thank you for the effort you put into this rebuttal. I appreciate the authors' detailed clarifications on some things I might have misunderstood but now I have more questions.
> >
> > > While the runtime varies based on the strength of the memorization, for verbatim memorized prompts, which represent the strongest type of memorization, the runtime was only a few seconds (as discussed in our limitations section in Appendix A).
> >
> > 1. Can you quantify the time taken to discover these memorization neurons? I would like to see the time taken to run Alg 1,2,3,4 separately.
> >
> > 2. I am a bit confused about the pipeline now as this is not explicitly stated in the paper. Do you (1) first find memorization neurons for every memorized prompt, and then prune the union of these memorization neurons from the model permanently? or (2) For every prompt, find memorized neurons, prune the model, evaluate, and then reset the model?

---

> > > ### Author Response · Authors · 2024-08-09
> > >
> > > Dear Reviewer zaL8,
> > >
> > > Thank you for your quick response and appreciating our rebuttal. We are happy to provide additional information and clarifications.
> > >
> > > 1.) We will run some additional experiments for your first question regarding the runtime of individual parts of our algorithm and provide them as soon as they become available.
> > >
> > > 2.) In our paper, we followed the first approach you sketched. We identify the neurons of each memorized prompt individually and then deactivate/prune the union of these neurons in the model. The metrics in the paper are computed on the U-Net, with all identified neurons removed simultaneously.

---

> > > ### Author Response · Authors · 2024-08-11
> > > **Additional Runtime Results**
> > >
> > > Dear Reviewer zaL8,
> > >
> > > Thank you for your question. We have rerun the runtime experiments and measured the time for each algorithm D1-D4. Please find the values in seconds in the following table:
> > >
> > > |    | D1   | D2   | D3   | D4    | Total |
> > > |----|------|------|------|-------|-------|
> > > | VM | 0.21 | 0.29 | 2.31 | 12.85 | 15.68 |
> > > | TM | 0.18 | 0.25 | 6.09 | 39.37 | 45.90 |
> > >
> > > In the table above, one can clearly see that the algorithms D1 and D2 take less than a second to run, while D3 takes roughly 2 seconds for verbatim memorized prompts and 6 seconds for template memorized prompts. As can be seen, D4 has the longest runtime. This is because each candidate neuron is checked individually for memorization by iterating over them and setting their output to zero individually. For template memorized samples, this takes a bit longer since, for this kind of memorization, our initial neuron candidate set is larger than for verbatim memorized samples. However, even though this process takes a bit longer for TM than for VM, the overall runtime is still only 45 seconds. Additionally, we want to emphasize here again that our algorithm only has to be run once. In contrast to existing mitigation approaches, once the memorization neurons are identified, the inference time, as well as the inference memory consumption, will be the same as that of the default diffusion model.

---

> > > > ### Comment · Reviewer_zaL8 · 2024-08-12
> > > > **Thank you for the rebuttal**
> > > >
> > > > Thank you for clarifying my doubts. I have a much better understanding of the paper. I appreciate that the authors calculated the time required to run their algorithm and I encourage them to add this analysis in the paper. I still have some concerns listed below.
> > > >
> > > > 1. The authors said that
> > > > >  While some sets of neurons memorize multiple prompts, no fixed set of neurons is responsible for memorizing all prompts. This is not a weakness of our method, but rather a reflection of the behavior of already trained models.
> > > >
> > > > This may be due to the method you proposed for searching neurons not being able to identify a fixed set of neurons. Previous studies have successfully localized memorization to a specific region of the model in image classifiers [1] and language models [2], suggesting that it could also be achievable in a diffusion model. The proposed method may lack the capability to accomplish this.
> > > > [1] Pratyush Maini, Michael C. Mozer, Hanie Sedghi, Zachary C. Lipton, J. Zico Kolter, and Chiyuan
> > > > 385 Zhang. Can neural network memorization be localized?
> > > > [2] Localizing Paragraph Memorization in Language Models, Stoehr et al
> > > >
> > > > 3. A significant limitation I see in the paper, which could be quite restricting, is that the proposed method requires observing all memorized prompts extracted for SD using a specific extraction method—in this case, the approach outlined in Wen et al.'s paper. However, this does not represent an exhaustive set of prompts that the model has memorized. Given that their method does not rely on identifying a fixed set of neurons, I am concerned that it may not extend to undiscovered prompts. The authors should consider explicitly stating this limitation in their paper as a disclaimer to avoid giving a false sense of security.
> > > >
> > > > Overall, I will lean toward a positive rating (and keep my rating) because I like the idea of localizing memorization in DMs and this paper presents interesting insights.

---

> ### Author Response · Authors · 2024-08-13
>
> Thank you for your prompt response and your inclination toward accepting our paper. We are still eager to convince you even more of our method.
>
> **1.)** We respectfully disagree with the concern that our method might not effectively identify the relevant neurons. For our set of memorized prompts, our method successfully identified 200 neurons in the first value layer out of a total of 320 neurons—constituting 62.5\% of the neurons in this layer. The memorization of prompts is actually distributed across multiple different neurons. Importantly, deactivating the neurons associated with the memorization of one prompt does not affect the memorization of other prompts, provided they do not share the same memorizing neurons. Furthermore, if memorization is triggered by neurons in the first value layer, deactivating neurons in subsequent layers does not prevent the generation of memorized samples. This observation extends to deeper layers as well—deactivating neurons in other value layers similarly does not mitigate memorization. Additionally, we have pinpointed neurons within the value layers of the U-Net’s first attention blocks as the primary trigger points of memorization, which represents a relatively small subset of the model's overall parameters.
>
> Regarding the hypothesis that a fixed set of neurons responsible for all memorization might exist, we would like to emphasize that the training objective of diffusion models (U-Nets) differs significantly from that of supervised image classifiers or next-token prediction in language models. Consequently, insights into memorization within these architectures may not directly apply to diffusion models. Without evidence or proof of a fixed set of neurons (which our experiments did not support), we believe this does not constitute a limitation of our method.
>
> We would also like to highlight that, similar to [1], "we determine the minimum number of neurons required to flip an example’s predictions" [1]. From our perspective, our method extends the line of research on memorization in supervised learning. The results we present for text-to-image models provide insights that are comparable to those from supervised learning, which identify a set of neurons for individual training images.
>
> **3.)** Our method does not necessarily rely on Wen et al.'s approach for detecting memorized prompts. As demonstrated in our previous response, our SSIM-based detection method is also capable of identifying memorization. Since it only requires a single denoising step, it can be applied at inference time to detect memorized prompts. We chose to use the evaluation dataset provided by Wen et al. to establish a benchmark and ensure our method is directly comparable to related work.
>
> Overall, we are very pleased that you appreciate our paper and the idea of localizing memorization in diffusion models. We will also expand our discussion in the paper to address the detection of previously undiscovered memorization. We thank you for this fruitful discussion.

---

### Official Review · Reviewer_Rr34 · 2024-07-12

**Soundness:** 3
**Presentation:** 4
**Contribution:** 4
**Rating:** 6
**Confidence:** 4

**Summary:**

This paper introduces NEMO, a method for localizing memorization in diffusion models (DMs) down to the level of individual neurons. The authors empirically evaluate NEMO on the Stable Diffusion model and demonstrate that deactivating the identified memorization neurons effectively mitigates memorization, increases image diversity, and maintains prompt alignment. The paper makes valuable contributions by providing insights into the memorization mechanisms in DMs and proposing a method for permanently mitigating memorization without compromising overall model performance or the quality of the generated images.

**Strengths:**

The paper presents an innovative and practical methodology, NEMO, for localizing memorization in DMs. This methodology allows model providers to identify and deactivate specific neurons responsible for memorizing training samples, effectively mitigating memorization.
The empirical findings presented in the paper provide valuable insights into the distribution of memorization neurons and the impact of deactivating these neurons on image generation. The experiments demonstrate that deactivating memorization neurons increases image diversity.
The paper is well-structured and provides a clear explanation of the proposed methodology, including the initial selection and refinement steps of NEMO. The authors also provide algorithmic descriptions for each component of NEMO in the appendix and supplement it with extensive experimental results.

**Weaknesses:**

The paper lacks a thorough evaluation of the proposed method's performance compared to existing baselines (if exist) or state-of-the-art methods for mitigating memorization in DMs. It would be beneficial to include comparisons with other methods to assess the novelty and effectiveness of NEMO.

**Questions:**

Can the authors provide more details on the selection of the memorization threshold $τ_mem$? How was it determined based on the pairwise SSIM scores? Did the authors consider any alternative methods for setting the threshold?
Have the authors considered comparing NEMO with other existing methods for mitigating memorization in DMs, such as rescaling attention logits or adjusting prompt embeddings? How does NEMO perform in terms of effectiveness and efficiency compared to these methods?

**Limitations:**

yes

---

> ### Author Rebuttal · Authors · 2024-08-06
>
> **Comparison to other mitigation methods?**
>
> > We have already quantitatively compared our method with the state-of-the-art mitigation approach by Wen et al. (ICLR 2024) in Section 4.1 and Table 1. Additionally, in Appendix C.9, we compared our method to the random token mitigation strategy proposed by Somepalli et al. (NeurIPS 2023), which achieved the best results among the mitigation methods they investigated. For the rebuttal, we also repeated the experiments using the attention scaling proposed by Ren et al. (Preprint 2024). We provide the additional results in Table 1 in the rebuttal PDF. Overall, our method not only performs comparably to Wen et al.'s approach in terms of mitigation and image-text alignment but also offers the significant advantage of precise memorization localization. This capability enables us to permanently eliminate memorization from the model after a single localization, thereby avoiding the need for additional repeated mitigation strategies during inference. Our approach is more cost-effective and computationally efficient for known memorized prompts than existing mitigation methods.
>
> > Conceptually, our method does not alter the input embeddings or attention score distribution; instead, it deactivates individual neurons in the U-Net. This approach eliminates the need for gradient computation, which can be memory-intensive. Notably, previous mitigation methods require adjustments and computations for each new generation. In contrast, once we have localized the memorizing neurons using our method, we can prune these neurons, eliminating the need for additional computations during future inference.
>
> **More details on the selection of the memorization threshold $\tau_{mem}$?**
>
> > We computed the threshold on a hold-out dataset of non-memorized prompts using 50,000 LAION prompts. By choosing a dataset of non-memorized prompts, we aimed to prevent overfitting our evaluation dataset of memorized prompts and thereby avoid biasing the results. We calculated the pairwise SSIM, our memorization score, between ten different initial noise samples for each of the 50,000 non-memorized LAION prompts and took the maximum score. We then averaged these memorization scores across all samples and added one standard deviation to set $\tau_{mem}$. This procedure is detailed in Section 3.1 of the paper. Since the SSIM score is only computed after the first denoising step, this threshold computation is computationally efficient.
>
> **How was $\tau_{mem}$ determined based on the pairwise SSIM scores?**
>
> > The lower we set the threshold, the more neurons will be deactivated since our algorithm aims to reduce the memorization score of the current sample (see Alg. 3 in Appx. D.3). We repeated our experiments with different choices of the threshold in Appx. C.7 to analyze the impact of different thresholds. A lower threshold identifies more neurons. However, for lower thresholds, the metrics are comparable to those obtained with our default threshold value in the main paper. Increasing the threshold reduces the number of identified neurons but slightly increases memorization, as measured by the SSCD scores. Thus, a trade-off exists between reducing the number of identified memorization neurons and their memorization mitigation effect. We also depict the impact of different threshold choices in Fig. 14 and Fig. 15 in Appx. C.7.
>
> **Alternative methods for setting the threshold $\tau_{mem}$**
>
> > We did not test other methods for setting the threshold. We believe that incorporating an additional hold-out set of memorized prompts might help users find even better thresholds. However, we focused on a practical scenario where the user only has access to a set of non-memorized prompts and demonstrated that this selection process is sufficient to localize and mitigate memorization in the model.

---

### Author Rebuttal · Authors · 2024-08-06

We thank all reviewers for their time spent reviewing our paper and their valuable feedback. We very much appreciate all the reviewers tend to accept the paper and think that the paper proposes an innovative and novel approach (Rr34, zaL8), presents interesting findings (zaL8, RyXV), provides a better understanding of the problem of memorization (K7EP), and is clearly written (Rr34, zaL8, K7EP). We address the individual questions and remarks below and are happy to provide further details if additional questions arise.

The attached PDF contains additional quantitative results for the attention scaling mitigation method proposed by Ren et al. (2024, Preprint) (Rr34), as well as images illustrating the impact of deactivating neurons in convolutional layers (K7EP) and the disentanglement of neuron activations in the value layers (RyXV).

We also emphasize that we have already provided a comprehensive comparison of our method to existing mitigation strategies in Table 1 of our main paper, along with additional results in Table 3 in Appendix C.3. An ablation study on the roles of individual key and value layers in memorization is presented in Appendix C.8. Furthermore, we explored the impact of different hyperparameters in our ablation and sensitivity analysis in Appendix C.7.

---

### Author Response · Authors · 2024-08-12
**Have concerns been addressed?**

Dear Reviewers,

We sincerely appreciate your questions and feedback, which have helped us to further improve our paper. We would like to follow up on our responses to ensure that they adequately address your concerns. If there are any additional questions or areas where further clarification is needed, please don't hesitate to reach out.

Thank you very much for your time and effort.

Best regards,
The Authors

---

### Decision · Program_Chairs · 2024-09-25

**Decision:**

Accept (poster)

**Comment:**

This paper proposed a simple method to localize memorization to individual neurons and showed that deactivating those neurons can have multiple benefits such as reduced memorization and increased diversity. The reviewers generally found this work interesting and novel. This would could have potential applications for future text-to-image generation model and system designs.